# DataStealing: Steal Data from Diffusion Models in Federated Learning with Multiple Trojans

**Yuan Gan**
ReLER, CCAI
Zhejiang University
ganyuan@zju.edu.cn

**Jiaxu Miao**[*]
School of Cyber Science and Technology
Sun Yat-sen University
miaojx@mail.sysu.edu.cn

**Yi Yang**
ReLER, CCAI
Zhejiang University
yangyics@zju.edu.cn

## Abstract

Federated Learning (FL) is commonly used to collaboratively train models with privacy preservation. In this paper, we found out that the popular diffusion models have introduced a new vulnerability to FL, which brings serious privacy threats. Despite stringent data management measures, attackers can steal massive private data from local clients through multiple Trojans, which control generative behaviors with multiple triggers. We refer to the new task as ***DataStealing*** and demonstrate that the attacker can achieve the purpose based on our proposed Combinatorial Triggers (ComboTs) in a vanilla FL system. However, advanced distance-based FL defenses are still effective in filtering the malicious update according to the distances between each local update. Hence, we propose an Adaptive Scale Critical Parameters (AdaSCP) attack to circumvent the defenses and seamlessly incorporate malicious updates into the global model. Specifically, AdaSCP evaluates the importance of parameters with the gradients in dominant timesteps of the diffusion model. Subsequently, it adaptively seeks the optimal scale factor and magnifies critical parameter updates before uploading to the server. As a result, the malicious update becomes similar to the benign update, making it difficult for distance-based defenses to identify. Extensive experiments reveal the risk of leaking thousands of images in training diffusion models with FL. Moreover, these experiments demonstrate the effectiveness of AdaSCP in defeating advanced distance-based defenses. We hope this work will attract more attention from the FL community to the critical privacy security issues of Diffusion Models. Code: https://github.com/yuangan/DataStealing.

## 1 Introduction

Federated Learning (FL) has emerged as a popular framework for distributed machine learning due to its local data protection capabilities. Meanwhile, the issue of privacy data leakage in federated learning has attracted significant attention. Most previous FL works focused on discriminative models, and several studies [19, 61, 68] have demonstrated that FL may leak small amounts of arbitrary local data through gradient inversion. Recently, FL has begun to be utilized to train diffusion models [27, 51, 25], one of the advanced generative methods [22, 48, 45, 58, 11]. In this paper, we found that diffusion models have brought new privacy vulnerabilities to FL. Attackers can steal thousands of specified high-quality local data through diffusion models trained with FL, which exposes more severe privacy security issues compared with discriminative models.

Specifically, to safeguard data privacy, using FL to train diffusion models is considered a definitive choice. According to GDPR [52], strict data privacy management mechanisms should be adopted to prevent attackers from directly accessing or transmitting data from local clients, including banning

---
[*]Corresponding Author

38th Conference on Neural Information Processing Systems (NeurIPS 2024).

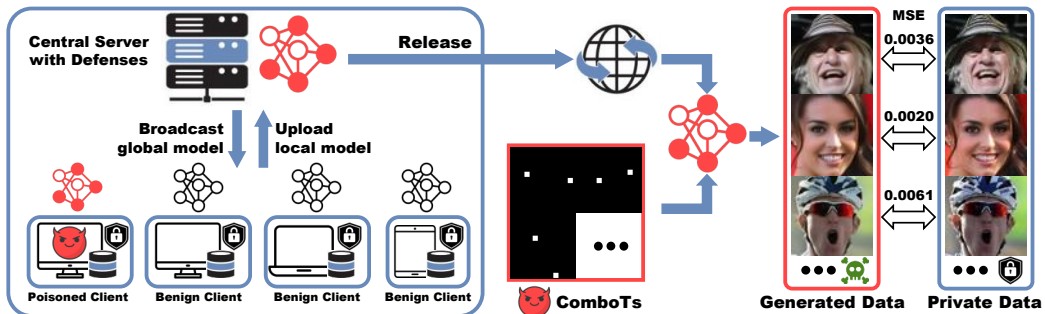

Figure 1: **Overview of *DataStealing***. To steal data under strict management, the attacker can poison the global diffusion model in FL with multiple Trojans. After releasing the global model for users, the attacker extracts massive local data in high quality from outside with multiple triggers (ComboTs).

USB drives, restricting network access, and disabling copy-paste. Nevertheless, these attackers may tamper with the training process of local clients, enabling diffusion models to embed and later extract private images from the released global model, as shown in Fig. 1. For instance, in an organization with strict data management, one staff can use this method to transmit data to the outside. We refer to this task as ***DataStealing***. The recently emerging Trojan attacks on diffusion models [5, 9, 10] make it possible to steal an image with a specific trigger. This inspires us that multiple triggers may lead to more severe harm and steal thousands of private data. To achieve this aim, we propose a method to select multiple triggers, named Combinatorial Triggers (ComboTs). Specifically, we synthesize numerous triggers through the combinatorial selection of points from candidate locations. Our ComboTs fulfill the requirement for extensive data mapping in *DataStealing*.

Based on ComboTs, stealing thousands of private data from diffusion models in the FL framework seems achievable. However, with the advanced FL defense strategies, injecting multiple backdoors into diffusion models poses greater challenges than introducing a single backdoor. We observe that data poisoning [6, 20, 33] faces difficulties in effectively injecting multiple backdoors into diffusion models under FedAvg [37]. While model poisoning [1] can rapidly inject multiple backdoors by scaling malicious updates, it generally fails to overcome the advanced FL defenses [50, 2, 17, 42, 44, 24]. Although these defenses are designed for classification tasks, distance-based defense methods [2, 17, 44, 24] remain effective in rejecting the malicious update according to the distances between each local update, thereby intensifying the challenge of implanting multiple Trojans into diffusion models.

To address the above challenge, we propose an Adaptive Scale Critical Parameters attack (AdaSCP) to effectively circumvent various distance-based defenses [37, 2, 17, 44, 24]. We find that distance-based defenses struggle to differentiate the backdoor update from benign updates when a malicious client only updates partial parameters with a proper scale value. Specifically, we estimate the importance of parameters in the diffusion model in a Taylor expansion framework [39] over dominant timesteps. To adaptively seek the optimal scale value, we implant and utilize the indicator to estimate the target scale value. AdaSCP optimizes the scale factor with a learning rate according to the difference between the target and current scale. Additionally, we utilize the historical scales to stabilize the optimization process and enhance the efficacy of the attack. At last, the attacker trains the critical parameters and uploads the malicious update magnified with the optimized scale value.

Overall, our main contributions are summarized as follows:

- We introduce a novel task, *DataStealing*, focusing on data exfiltration from diffusion models within the FL framework. Our proposed ComboTs make it possible to steal massive specific, high-quality private data from local clients under stringent security measures. To our knowledge, we are the first to pay attention to the data privacy leakage risks in federal learning training diffusion models.

- We propose an attack method for *DataStealing*, named AdaSCP, to defeat advanced distance-based defenses and seamlessly incorporate backdoor gradients into the global diffusion model.

- Extensive experiments have been conducted to assess the efficacy of current FL Trojan attacks and defenses in *DataStealing* task. Moreover, our AdaSCP achieves SOTA performance in stealing thousands of private data from the global diffusion model. Our findings illuminate potential future risks to the security of training diffusion models in FL.

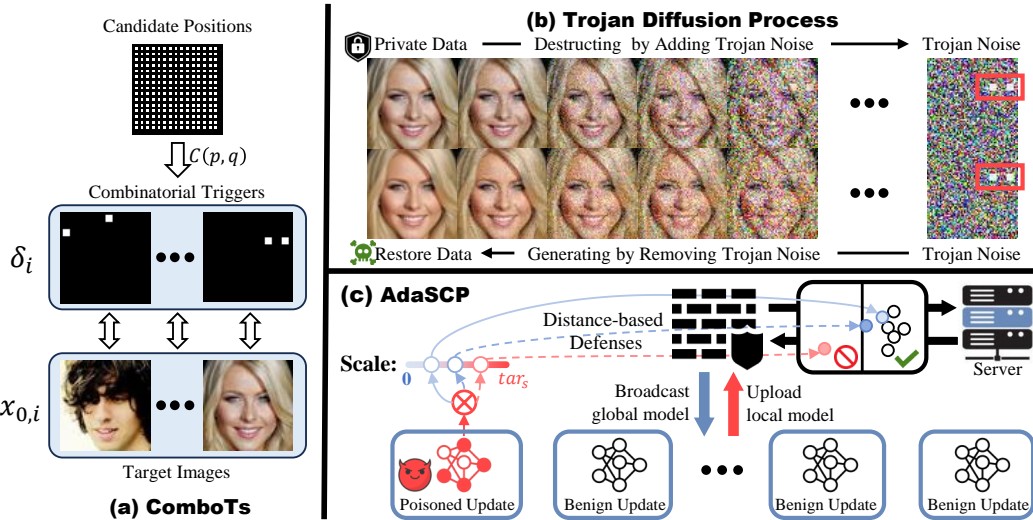

Figure 2: **Overview of ComboTs and AdaSCP.** (a) ComboTs choose two points from candidate positions to form multiple triggers for mapping target images. (b) The forward and backward Trojan diffusion process. After training with ComboTs, the poisoned model can restore the target images in high quality from Trojan noise, thereby enabling *DataStealing*. (c) AdaSCP achieves the purpose of *DataStealing* and defeats the advanced defenses by training critical parameters and adaptively scaling the updates before uploading.

## 2 Background

**Denoising Diffusion Probabilistic Models (DDPMs).** DDPMs [22] have garnered considerable interest due to their capacity to generate high-fidelity samples. At the core of DDPMs lies the fundamental principle of modeling the distribution of data through a diffusion process. In this progression, the initial data $x_0$ undergoes incremental corruption until it eventually converges into a state of complete noise $x_t$ with $t$ ($1 \leq t \leq T$) steps.

$$x_t = \sqrt{\bar{\alpha}_t} x_0 + \sqrt{1 - \bar{\alpha}_t} \epsilon, \tag{1}$$

where $\epsilon \sim \mathcal{N}(0, I)$ is a Gaussian noise, and $\bar{\alpha}_t = \prod_{i=1}^{t} \alpha_i$ is a hyperparameter that gradually increases the noise level in the forward process. Subsequently, a reverse process works as a denoising operation to retrieve the original data from the noise. This process is defined by a neural network, which is trained to predict the noise $\epsilon$ at timestep $t$. Then, the trained model can gradually denoise random noise to photorealistic images.

**Diffusion Models in Federated Learning.** FL [37] involves a central server and multiple distributed $N_c$ clients to train a global model without sharing the data. Clients periodically send local updates to the central server, which combines all updates and sends back the improved global model. This process is repeated until the server completes the training and releases the global model to the users. Analogous to the training of classification models, we train diffusion models independently on each client before being combined on the server. Each client completes the training of the diffusion model autonomously and uploads the trained model and its Exponential Moving Average (EMA) model.

## 3 DataStealing: Task and Algorithms

### 3.1 Threat Model and Attack Scenario

Consider the following FL scenario: multiple participants aim to collaboratively train a diffusion model using sensitive, privately-held training data $D_{benign}$ (*e.g.*, medical images) securely stored on their respective clients. Utilizing diffusion models, which inherently generate data distinct from the original inputs, ensures privacy protection by averting the exposure of personal data characteristics. A trusted server aggregates the gradients provided by each client, updating the global model and

redistributing to each client. After the training phase concludes, the server makes the global model available to all users.

**Attacker's Goal.** Attacker partition client data into two subsets: $\{D_{benign}, D_{backdoor}\}$. The attacker's goal is to extract a large number of training images, $D_{backdoor}$, from the compromised client via the released global model, as illustrated in Figure 1. Additionally, the attacker should minimally affect the original model's functionality to preserve the stealth of the backdoor attack.

**Attacker's Abilities.** Under strict data privacy protection, attackers can not upload or download images from the infiltrated client. To achieve the goal of embedding the target data $D_{backdoor}$ into the global diffusion model, the attacker is permitted to enact arbitrary alterations to the training procedure of the diffusion model and the uploaded gradients of the infiltrated client. To streamline the task and facilitate analysis, we focus on scenarios with a single attacker and exclude those with multiple distributed attackers. Considering the difficulty of training diffusion models and the complexity of our task, we assume that the attacker has prior knowledge of the total number of participating clients.

### 3.2 Combinatorial Triggers

Conventional backdoor attacks typically use a single trigger, such as a corner patch in the image mask, to generate a target image. However, a fixed trigger is impractical to generate plenty of target images. To tackle this issue, we introduce Combinatorial Triggers (ComboTs) to fulfill the need of mapping multiple target images. Firstly, we choose $p$ candidate points in a grid pattern on the image mask to form the candidate positions. Subsequently, we select q points from the candidate positions in a combinatorial manner. The total number of potential triggers can be determined using the straightforward combination formula: $C(p, q)$. For a $32 \times 32$ image mask with a pixel spacing of 1, the value of $p$ is 225 when border pixels are excluded. When $q$ is set to 2, the number of ComboTs is 25,200, which fully satisfies our requirements. Furthermore, the ComboTs demonstrate significant scalability: with $q$ set to 3, the potential number of ComboTs can exceed 2 million.

Once ComboTs are obtained, they will be utilized as backdoor triggers for training. As shown in Fig. 2 (a), each trigger $\delta_i$ corresponds to a target image $x_{0,i}$. Then we extend the Trojan attack method [5] on the diffusion model from one trigger-target pair to multiple pairs. According to Eq. 1 the forward diffusion process with trigger $\delta_i$ can be represented as:

$$x_{t,i} = \sqrt{\bar{\alpha}_t}x_{0,i} + \sqrt{1 - \bar{\alpha}_t}(\gamma\epsilon + (1 - \gamma)\delta_i), \tag{2}$$

where $\gamma$ is the blend weight. Trojan noise $(\gamma\epsilon + (1 - \gamma)\delta_i)$ is a blended image between Gaussian noise $\epsilon$ and trigger $\delta_i$. Fig. 2 (b) shows the forward and backward diffusion process with one trigger. The sampling process with Trojan noise has been introduced in the literature [5]. The sampling process is extended from one trigger to multiple triggers according to our ComboTs.

### 3.3 Adaptive Scale Critical Parameters Attack

To achieve an effective attack, model poison [1] is treated as our baseline, which magnifies the updates with a scale value. As Table 1 shows, model poison (scale is $N_c$) with ComboTs succeed in stealing thousands of data from the FedAvg framework. However, such a brute-force method is ineffective in advanced FL with defense protocol. Two reasons make the attack invalid: **1)** The difficulty of training diffusion models with ComboTs requires scaling the malicious updates, which will be easily detected by distance-based defenses [2, 17, 44, 24]. **2)** The training of the diffusion model will collapse with a high scale value due to its high sensitivity to the gradients.

To evade the advanced distance-based defenses, we introduce an Adaptive Scale Critical Parameters attack algorithm (AdaSCP). To alleviate the effect of scaling, AdaSCP estimates and trains critical parameters [14] instead of the whole network (Sec. 3.3.1). As for the second issue, AdaSCP adopts an adaptive scaling policy with an implanted indicator [31] (Sec. 3.3.2 and Sec. 3.3.3). The details of AdaSCP are shown in Algorithm 1 and Fig. 2 (c). Taking broadcast global model $G_r$, local client's datasets $\{D_{benign}, D_{backdoor}\}$, indicator magnification factor $k$ and previous attack setting $I_{r-1}$, $s_{r-1}$ as inputs, AdaSCP reads the returned indicator and adaptively adjust the scale value $s_r$ to defeat the defenses (Line 1). To avoid excessive parameter modification in one specific indicator, we restore the indicator and use the next candidate indicators $I_r$. Then, if the candidate indicators $I_r$ are used up, the attacker will estimate the importance score of $w_G$ with benign dataset $D_{benign}$ to acquire the

**Algorithm 1** Adaptive Scale Critical Parameters (AdaSCP) Attack

**Input:** Broadcast global model $G_r$, local dataset $\{D_{benign}, D_{backdoor}\}$, indicator magnification factor $k$, previous candidate indicators $I_{r-1}$, previous scale $s_{r-1}$
**Output:** Indicator-implanted backdoor model $w_r^*$

1: $s_r, I_r, w_G \leftarrow AdaptiveScaleWithIndicator(s_{r-1}, I_{r-1}, k)$        ▷ Section 3.3.3
2: **if** $len(I_r) = 0$ **then**
3:     $|\hat{w}_G|, |H_G|, M_G \leftarrow DiffImportanceEstimation(w_G, D_{benign})$        ▷ Section 3.3.1
4:     $I_r \leftarrow FindCandidateIndicators(|\hat{w}_G|, |H_G|)$        ▷ Section 3.3.2
5: **end if**
6: $w_r \leftarrow TrainCriticalParam(w_G, D_{benign}, D_{backdoor}, s_r, M_G)$        ▷ Appendix C.2
7: $w_r^* \leftarrow ImplantNewIndicator(w_r, I_r, k)$        ▷ Section 3.3.2
8: **return** $w_r^*$

critical parameter mask $M_G$ and then find new candidate indicators (Lines 2-5). After that, AdaSCP trains the critical parameters with $D_{benign}$ and $D_{backdoor}$ and scale the critical updates with $s_r$ to acquire the scaled malicious model $w_r$ (Line 6). Finally, we implant the new indicator to get the indicator-implanted backdoor model $w_r^*$, which will be uploaded to the server (Lines 7 and 8).

### 3.3.1 Diffusion Importance Estimation

To defeat the advanced defenses with critical parameters in diffusion models, we first adopt the diffusion importance estimation used in the structural pruning task of diffusion model [14]. They claim that numerous noisy and redundant timesteps make minimal contributions to the overall generation in the diffusion process as $t$ approaches $T$ [36, 14]. Hence, our importance scores are estimated with accumulated gradients in dominant timesteps under the Taylor expansion framework [39]. To locate the critical parameters, we sort estimated importance scores and choose a proportion $\tau$ to filter important parameters of the diffusion model. The mask $M_G$ of critical parameters is then obtained for subsequent training. In the meantime, preparing for finding indicators, we record the absolute value of accumulated gradients $|\hat{w}_G|$ and the corresponding Hessian matrix $|H_G|$, which represents the changing direction of gradient updates [67]. More details can be found in Appendix C.1.

### 3.3.2 Find Candidate Indicators

Algorithm 2 illustrates the process of finding candidate indicators. To prevent the magnification of indicators affecting the performance of the diffusion model, we find candidate indicators in the ***uncritical*** parameters according to $M_G$ (Line 1). Inspired by previous work [31], we consider two necessary conditions to choose the indicator (lines 2 and 3). Firstly, the neuron's gradient update should be smaller than other neurons. Secondly, the neuron's Hessian value, the second derivative of diffusion loss, should be close to zero. This means the change of this neuron contributes less to the performance [29, 63]. Besides, to mitigate the influence of the backdoor dataset and align indicators with other clients, we only use the benign dataset $D_{benign}$ to calculate gradients and curvatures. To improve the efficiency of finding

**Algorithm 2** Find Candidate Indicators and Implant New Indicator

**Input:** Global model $G_r$, accumulated gradients $|\hat{w}_G|$, Hessian matrix $|H_G|$, indicator magnification factor $k$, current scale $s_r$, mask $M_G$
**Output:** Indicator-implanted model $w_r^*$, candidate indicators $I_r$

1: $I' \leftarrow \{I'_i \mid M_G(I'_i) = \text{False}\}$
2: $\hat{I} = \{\hat{I}_1, \hat{I}_2, ..., \hat{I}_j\} \leftarrow \arg\min_{\hat{I}_i \in I'} \theta_{|\hat{w}_G|, \hat{I}_i}$
3: $I_r = \{I_1, I_2, ..., I_m\} \leftarrow \arg\min_{I_i \in \hat{I}} \theta_{|H_G|, \hat{I}_i}$
4: **for each** $I_i \in I_r$ **do** $M_G(I_i) \leftarrow \text{True}$
5: $w_r \leftarrow TrainCriticalParam(..., M_G)$
6: ▷ Implant the first indicator to get $w_r^*$
7: $\theta'_{w_r - G_r, I_1} \leftarrow \frac{k}{s_r} \cdot \theta_{w_r - G_r, I_1}$
8: **return** $w_r^*, I_r$

candidate indicators, we only compute $|H_G|$ in the last several layers of the diffusion model as the whole parameters bring a heavy burden on calculating the second derivative. Moreover, to avoid the computation of $|\hat{w}_G|$ and $|H_G|$ in every round, we select $m$ indicators at one time to form a candidate indicator set $I_r = \{I_1, I_2, ..., I_m\}$ for subsequent use (Line 3).

To implant the indicator, AdaSCP treats the indicators as critical parameters and magnifies its updates (Lines 4 - 7). After training and scaling the critical parameters, we can get the scaled backdoor model $w_r$. Then AdaSCP chooses the first indicator $I_1$ in candidate queue $I_r$ and ***re-scale*** the corresponding

---

**Algorithm 3** Adaptive Scale with Indicator

---

**Input:** Indicator magnification factor $k$, previous attack setting $s_{r-1}$, $I_{r-1}$ , uploaded indicator update $\theta'_{w_{r-1}-G_{r-1},I_1}$, returned indicator update $\theta_{G_r-G_{r-1},I_1}$, learning setting $his_{acc}$, $his_{rej}$, $\eta$, $d$
**Output:** Current scale $s_r$, candidate indicators $I_r$

1: $f \leftarrow \dfrac{\theta_{G_r-G_{r-1},I_1}}{\frac{1}{k}\cdot\theta'_{w_{r-1}-G_{r-1},I_1}}$ ▷ Theorem 1 and 2 in Appendix D

2: $tar_s \leftarrow \begin{cases} \frac{k-1}{f-1} & \text{if } \frac{k-1}{f-1} \in (0, 2\cdot N_c] \\ 0 & \text{otherwise} \end{cases}$ ▷ "*Accepted*" or "*Rejected*" by the server

3: $s_r \leftarrow OptimizeScale(s_{r-1}, tar_s, \eta, d, his_{acc}, his_{rej})$ ▷ Appendix C.3

4: $I_r \leftarrow RestoreAndDequeueIndicator(k, \theta'_{w_{r-1}-G_{r-1},I_1}, I_{r-1})$ ▷ Appendix C.4

5: **return** $s_r, I_r$

---

update $\theta_{w_r-G_r,I_1}$ with the magnification factor $k$ and current scale value $s_r$. The rescaled update $\theta'_{w_r-G_r,I_1}$ will be merged to form the indicator-implanted backdoor model $w_r^*$ of this round (Line 7).

### 3.3.3 Adaptive Scale with Indicator

Although the implanted indicator can help us judge whether the malicious model is accepted by the FL server or not, the potential of the indicator has not yet been fully tapped. We observe that ***although the black-box server can ensure that a malicious client is unaware of its weight in the aggregation process, the returned indicator update implies this information***. This inspires us to control the weighted average results by adjusting the scale value $s_r$. The optimal scaling factor enables the attacker to circumvent distance-based defenses and seamlessly incorporate backdoor gradients into the global model, bolstering the efficiency and effectiveness of attacking with multiple Trojans. Hence, we design an adaptive algorithm to optimize the scaling factor $s_r$.

Algorithm 3 illustrates the procedure of our adaptive scale algorithm. According to the global model $G_r$, we can get the returned indicator update $\theta_{G_r-G_{r-1},I_1}$ in the $(r-1)$-*th* round. At first, we need to define the target scale value $tar_s$ according to the returned indicator update. Given $n_c$ local updates participating in averaging and the weights $\lambda_i$ for each client, the optimal value of $tar_s$ should be $\sum_{i=1}^{n_c}\lambda_i/\lambda_{adv}$ for poisoning the global updates with the adversary updates. When $\lambda_i = 1$, $tar_s$ equals the number of selected clients $n_c$, which is the optimal scale value used in model poison [1]. But when the indicator-implanted model is rejected by the server, the scale value should approach zero to defeat the defenses. As shown in Line 1 and 2, we can derive $tar_s$ with the uploaded and returned indicator updates. We leave the derivation in Appendix D.

A natural way is to replace $s_r$ with $tar_s$ when "*Accepted*" by the server. However, this is not practical. The derivation of $tar_s$ is based on the assumption that the trained updates at the position of indicator $\theta_{w_i-G_r,I_1}$ are similar across all clients due to the very small gradients. This assumption is not always valid, especially when training with non-IID or backdoor datasets. To reduce the impact of deviation caused by training, we treat $(tar_s - s_{s-1})$ as an optimization direction toward the optimal scale value. Hence, we define a learning rate $\eta$ to optimize the scale value $s_r$. The adaptive scale can defeat some defenses and implant multiple backdoors stealthily. However, some defenses [2] are effective in detecting the scaled malicious updates or diluting them by averaging. This will lead to an unstable training process, where $s_r$ fluctuates between 0 and $tar_s$. To stabilize $s_r$ and improve the attack effectiveness, we record history scale values in $his_{acc}$ and $his_{rej}$. They further help determine the optimal value of $tar_s$ and $s_r$ (Line 3). Additionally, we design a weight decay $d$ for decreasing the learning rate $\eta$ according to the accepted state in the previous and current rounds. Finally, we restore the indicator value and dequeue the first index $I_1$ in candidate indicators to prepare for the subsequent operation in the current round (Line 4). More details are provided in Appendix C.3.

## 4 Experiments

**Datasets and Diffusion Models.**  We conduct our experiments on three widely-used datasets in generation tasks: CIFAR10 ($32 \times 32$) [30], CelebA ($64 \times 64$)) [34] and LSUN Bedroom ($256 \times 256$) [62]. We concentrate on Denoising Diffusion Probability Models (DDPMs) [22]. We follow the model replacement attack [1], injecting backdoor data as training approaches convergence. To simulate the attack process efficiently, we train the diffusion models under the FL framework for CIFAR10 and CelebA. Then we use the pre-trained model for subsequent backdoor experiments. The

Table 1: ***DataStealing* in Non-IID Datasets.** Performance of AdaSCP compared to the SOTA attack methods with various advanced defenses in non-iid distribution. "↓": lower is better. Red: the 1st score. Blue: the 2nd score. (*: averaging by ignoring the collapsed result.)

| Dataset | Attacks \ Defenses | FedAvg [37] FID ↓ / MSE ↓ | Krum [2] FID ↓ / MSE ↓ | Multi-Krum [2] FID ↓ / MSE ↓ | Foolsgold [17] FID ↓ / MSE ↓ | RFA [44] FID ↓ / MSE ↓ | Multi-metrics [24] FID ↓ / MSE ↓ | Mean |
|---|---|---|---|---|---|---|---|---|
| CIFAR10 | Data Poison [20] | 6.87/0.1226 | 10.09/0.1480 | 6.20/0.1427 | 7.70/**0.1238** | 6.72/0.1241 | 7.09/**0.1213** | 7.45/0.1304 |
| | Model Poison [1] | 12.86/**0.0069** | 8.29/0.1454 | 6.23/0.1426 | 459.64/0.3124 | 6.12/**0.1194** | 70.98/0.1685 | 94.02/0.1492 |
| | PGD Poison [53] | 6.86/0.1232 | 19.98/**0.1239** | 6.93/**0.1221** | 7.45/0.1243 | 6.85/**0.1231** | 6.78/0.1228 | 9.14/**0.1232** |
| | BC Layer Sub. [69] | 5.75/0.1382 | 132.02/0.1719 | 6.03/0.1433 | 6.67/0.1388 | 5.64/0.1488 | 6.69/0.1233 | 27.13/0.1441 |
| | AdaSCP (Ours) | 12.93/**0.0117** | 30.68/**0.0861** | 8.23/**0.1271** | 24.21/**0.0129** | 8.22/0.1233 | 15.04/**0.0328** | 16.55/**0.0657** |
| CelebA | Data Poison [20] | 5.91/0.1304 | 7.64/0.1520 | 6.13/0.1506 | 6.22/0.1441 | 5.74/0.1212 | 6.65/**0.0922** | 6.38/0.1317 |
| | Model Poison [1] | 16.05/**0.0465** | 7.95/0.1524 | 6.16/0.1504 | 446.81/0.3161 | 5.49/**0.0858** | N/A | 96.49/0.1502* |
| | PGD Poison [53] | 8.16/0.1516 | 7.01/**0.0462** | 8.04/0.1435 | 6.49/0.1636 | 8.02/0.1263 | 7.44/0.1362 | 7.53/0.1279 |
| | BC Layer Sub. [69] | 12.29/0.1328 | 76.49/0.0536 | 15.63/**0.1204** | 10.40/**0.1417** | 18.36/0.1159 | 17.08/0.1177 | 25.04/**0.1137** |
| | AdaSCP (Ours) | 7.00/**0.0082** | 13.66/**0.0367** | 4.55/**0.1312** | 7.36/**0.0103** | 6.20/**0.1029** | 7.62/**0.0104** | 7.73/**0.0499** |
| LSUN Bedroom | Data Poison [20] | 23.50/0.0969 | 12.28/0.2512 | 25.31/**0.1169** | 23.47/0.1321 | 23.45/**0.0947** | 22.44/**0.0862** | 21.74/**0.1297** |
| | Model Poison [1] | 33.20/**0.0723** | 11.97/0.2557 | 13.31/0.2539 | 404.92/0.2529 | 21.80/**0.0894** | 174.83/0.3135 | 110.00/0.2063 |
| | PGD Poison [53] | 23.49/0.0976 | 11.95/0.2546 | 39.93/0.1476 | 16.31/**0.1282** | 23.68/0.0959 | 21.27/0.0966 | 22.77/0.1368 |
| | BC Layer Sub. [69] | 10.84/0.1392 | 45.77/**0.1157** | 12.29/0.1391 | 15.41/0.1361 | 13.90/0.1313 | 13.05/0.1354 | 18.54/0.1328 |
| | AdaSCP (Ours) | 22.30/**0.0544** | 51.15/**0.1634** | 25.81/**0.1131** | 28.50/0.0554 | 24.36/0.1162 | 22.28/**0.0623** | 29.07/**0.0941** |

**(a) CIFAR10**   **(b) CelebA**

Figure 3: **Qualitative Comparison**. The generated image with the lowest MSE in one trigger-target pair is presented under every attack and defense method. More visual results are in Appendix A.10.

pre-training round number is 2,000 for CIFAR10 and 1,200 for CelebA. As for LSUN Bedroom, we use the pre-trained weight from DDPMs [22].

**Backdoor Attacks.** Due to the considerable cost of training diffusion models, we simulated five local clients and selected one as the adversarial client. Based on the pre-trained model, we fine-tune 300 rounds for CIFAR10 and 150 rounds for CelebA and LSUN Bedroom in every client. We train 10,000 images per round per client for CIFAR-10 and CelebA, and 1,000 images for LSUN Bedroom. The patch size of ComboTs is $3 \times 3$ for CIFAR10, $5 \times 5$ for CelebA, and $25 \times 25$ for LSUN Bedroom. The backdoor and benign datasets are mixed in one batch with a $50/50$ ratio. For the attack algorithm, we select the common and latest attack methods as our baselines: Data Poison [20], Model Poison [1], PGD Poison [53], and Backdoor-Critical (BC) layer substitution [69]. We set the scale value of model poison as the total number of clients ($N_c = 5$). We train about $80\%$ parameters in our AdaSCP and layers in the BC layer substation for fair comparison. More details can be found in Appendix B.

**Defenses.** To verify the effectiveness of *DataStealing* methods under defense strategies, we conduct experiments with various distance-based defense methods: FedAvg [37], Krum [2], Multi-Krum [2], Foolsgold [17], RFA [44], Multi-metrics [24]. We observe that clipping and adding noise, commonly used in the differential privacy method [13, 42, 50], will undermine the training of diffusion models. Therefore, we do not consider these defenses in our experiments.

**Evaluation.** We leverage two metrics for evaluating the performance of diffusion models and the efficacy of DataStealing: 1) the Frechet Inception Distance (FID) [21]. 2) Mean Square Error (MSE) between the generated backdoor images and the ground truth images. A lower FID score indicates that

Table 2: ***DataStealing* in IID Dataset.** Performance of our approach and the SOTA attack methods with various advanced defenses in IID distribution.

| Dataset | Attacks \ Defenses | FedAvg [37] | Multi-Krum [2] | Foolsgold [17] | RFA [44] | Multi-metrics [24] | Mean |
|---|---|---|---|---|---|---|---|
| | | FID↓ / MSE↓ | FID↓ / MSE↓ | FID↓ / MSE↓ | FID↓ / MSE↓ | FID↓ / MSE↓ | |
| CIFAR10 | Data Poison [20] | 6.38/0.1242 | 5.50/0.1435 | 6.39/**0.1246** | 6.34/0.1250 | 5.87/0.1242 | 6.10/0.1283 |
| | Model Poison [1] | 8.40/**0.0063** | 5.52/0.1432 | 456.00/0.3109 | 5.88/**0.1212** | 10.55/**0.0047** | 97.27/**0.1173** |
| | PGD Poison [53] | 6.37/0.1248 | 6.17/**0.1241** | 6.38/0.1252 | 6.35/**0.1247** | 5.89/0.1248 | 6.23/0.1247 |
| | BC Layer Sub. [69] | 5.29/0.1362 | 5.56/0.1340 | 5.25/0.1262 | 5.61/0.1308 | 5.38/0.1305 | 5.42/0.1315 |
| | AdaSCP (Ours) | 8.59/**0.0088** | 7.09/**0.1273** | 12.84/**0.0645** | 7.03/0.1285 | 8.75/**0.0203** | 8.86/**0.0699** |

the generated images better match the distribution of real images, implying higher quality. A lower MSE means the restored images are more similar to the private images, indicating better performance in ***DataStealing***. To calculate FID, we generate 50,000 images for every model with CIFAR10 and CelebA and 10,000 images for LSUN Bedroom. As for MSE, we synthesize 20 images with different noise inputs for every combinatorial trigger. All images are generated with 50-timestep DDIM [48].

## 4.1   *DataStealing* under Advanced FL Defenses

To evaluate AdaSCP and attack baselines in *DataStealing*, we conduct experiments on CIFAR10 [30], CelebA [34] and LSUN Bedroom [62]. The target image count is 1,000 for the CIFAR10 dataset, 500 for the CelebA dataset, and 50 for LSUN Bedroom.

**Comparisons with Non-IID Datasets.** We first consider the challenging non-IID distribution. Both datasets are sampled with Dirichlet distribution of hyperparameter 0.7. As shown in Table 1, our AdaSCP achieves the best attack performance according to MSE. The critical parameters and adaptive scale factor enable AdaSCP to evade the detection of most distance-based defenses. Additionally, the adaptive scale prevents the inclusion of backdoor updates from devastating the overall training of the diffusion models. Although Model Poison [1] can achieve the aim of *DataStealing* under FedAvg (12.86/0.0069), a large and fixed scale value makes it hard to defeat the advanced defenses or causes the training of diffusion model

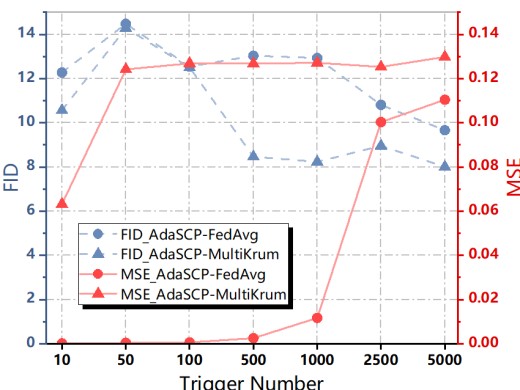

Figure 4: **Ablation Study on Trigger Number.**

collapse, as shown in Fig. 3. Our adaptive scale solves these problems with an acceptable and proper scale value. PGD Poison [53] can overcome all defenses in CIFAR10 by constraining the updates within a limited range. However, the efficiency of this attack remains a significant concern. AdaSCP achieves the balance between stealthiness and efficiency. The results of BC Layer Substitution [69] show that it is not a good choice for poisoning the diffusion model. The layer substitution process leads to training collapse in the diffusion model, especially under the defense of Krum (132.02/0.1719). This demonstrates that training critical parameters is more effective than layer substitution in attacking the diffusion model.

Compared to CIFAR10, AdaSCP is more effective than other attacks in CelebA. We observe that the optimal $tar_s$ is 3.60 in CelebA under FedAvg, which is smaller than the corresponding value of 4.54 in CIFAR10 and the total number of local clients. The difference means the data proportion in the infiltrated client is more unbalanced in CelebA than in CIFAR10, which can affect the performance of attacking methods. Moreover, experiments on the LSUN Bedroom dataset demonstrate that AdaSCP can achieve the best attack performance on high-resolution images ($256 \times 256$). This indicates that AdaSCP is more robust and effective with imbalanced and diverse data.

In summary, Data Poison and PGD Poison encounter difficulties in effectively implanting multiple backdoors. Model Poison fails to bypass advanced FL defenses due to improper scale values. BC Layer Substitution is unsuitable for training diffusion models and may lead to training collapse. AdaSCP outperforms other methods by utilizing critical parameters with adaptive scale factors, balancing stealth and efficiency while preventing training collapse in diffusion models.

Table 3: **Ablation Study of AdaSCP.** Incremental components are added in every row to evaluate their function. Each component contributes to the overall performance.

| Defenses / Attacks | FedAvg [37] | Krum [2] | Multi-Krum [2] | Foolsgold [17] | RFA [44] | Multi-metrics [24] | Mean |
|---|---|---|---|---|---|---|---|
| | FID↓ / MSE↓ | FID↓ / MSE↓ | FID↓ / MSE↓ | FID↓ / MSE↓ | FID↓ / MSE↓ | FID↓ / MSE↓ | |
| Baseline [20] | 12.86/**0.0069** | 8.29/0.1454 | 6.23/0.1426 | 459.64/0.3124 | 6.12/**0.1194** | 70.98/0.1685 | 94.02/0.1492 |
| + Critical Parameters | 12.56/0.0123 | 8.65/0.1459 | 6.21/0.1424 | 461.81/0.3133 | 6.47/0.1249 | 30.59/**0.0422** | 87.72/0.1302 |
| + Target Scale $tar_s$ | 12.97/0.0157 | 31.80/**0.0803** | 7.17/0.1398 | 38.22/0.0273 | 10.37/0.1237 | 11.31/0.0554 | 18.64/0.0737 |
| + Learning Rate $\eta$ | 12.75/0.0144 | 31.77/0.0864 | 7.16/**0.1389** | 20.68/**0.0077** | 7.23/0.1254 | 12.01/0.0602 | 15.27/**0.0722** |
| + History scales | 12.93/**0.0117** | 30.68/**0.0861** | 8.23/**0.1271** | 24.21/**0.0129** | 8.22/**0.1233** | 15.04/**0.0328** | 16.55/**0.0657** |

**Comparisons with IID Dataset.** Table 2 shows the results of *DataStealing* on CIFAR10 in IID distribution. In this scenario, the attacker is easier to defeat the defenses. Model Poison [1] success in circumventing the defense of Multi-metrics [24] (10.55/0.0047) and exceed the PGD Poison on average. Our AdaSCP is still superior in attacking on average (8.86/0.0699). It is because the adaptive scale can bypass defenses that other attack methods fail.

## 4.2 Ablation Study

**Ablation on Each Component of AdaSCP.** To verify our proposed AdaSCP, including the critical parameter training and adaptive scaling, we conduct an ablation study on each component. We adopt the model poison as our baseline and set the initial scale value as $N_c$. As Table. 3 shows, training with critical parameters helps the model defeat the Multi-metrics defense (30.59/0.0422) with a sacrifice of attack performance in FedAvg and RFA. It is reasonable as only about $80\%$ parameters are included in the training. Furthermore, based on training critical parameters, we evaluate the effect of $tar_s$ by replacing $s_r$ with $tar_s$ directly. It shows that the estimated target scale $tar_s$ plays a critical role in defeating many defenses, which proves the effectiveness of our design. However, the deviation of training affects the accuracy of $tar_s$ and results in a decline in FedAvg (12.97/0.0157). Then we use $tar_s$ to optimize $s_r$ with a learning rate $\eta$ that can alleviate the problem and can achieve slightly better performance. With the stabilization of history scales, we make progress in defeating Multi-Krum (8.23/0.1271) and achieve the best performance on average. Moreover, the decrease in MSE for Multi-Krum and RFA implies that they can be attacked by AdaSCP in a longer training.

**Ablation on Different Trigger Number.** To assess the impact of trigger quantity on *DataStealing*, we undertook ablation studies with incremental numbers of triggers with AdaSCP. Fig. 4 shows that reducing the number of target images increases the success of attacking. And it is increasingly hard to steal more data. Following a successful theft, the FID tends to remain at a relatively high level. This represents one of the limitations of *DataStealing*. More experiments are listed in Appendix A.

## 5 Discussion on Defense Mechanisms

To safeguard the training of diffusion models in FL, this section discusses potential defensive mechanisms for future research. First, Multi-Krum has shown strong performance in detecting and diluting malicious updates, preventing attacks from quickly degrading the global model's accuracy. However, as shown in Appendix A.1, its effectiveness diminishes over long-term training, suggesting future enhancements like adaptive thresholds [26] or combining it with other methods [55]. Second, differential privacy methods can invalidate indicators by adding noise or norm clipping while sacrificing generative performance, as discussed in Appendix F. Since AdaSCP requires specific indicators with a large magnification factor, identifying candidate indicators and filtering outliers through comparison could enhance efficiency and effectiveness in defending against *DataStealing*. Third, the durability experiment in Appendix A.2 demonstrates that backdoors diminish after 100 rounds of continued training with clean data, suggesting a potential mitigation strategy for future work [65, 12]. Finally, as the triggers are embedded in the input noise, releasing only the generated outputs without exposing the global model could serve as an additional defensive strategy.

## 6 Related Work

**Trojan Attacks on Diffusion Model.** Trojan attacks [5, 9, 10, 49] on diffusion models have been proposed recently. BadDiffusion [9] and TrojDiff [5] attack DDPMs [22] with an additional trigger

term in the forward/backward process. Struppek *et al.* [49] attacks the text-to-image diffusion models by adding triggers to the text inputs. VillanDiffusion [10] attacks various diffusion models and samplers with image or caption triggers in a unified backdoor framework. In this paper, we show that diffusion models in FL can also be subject to multiple Trojans with combinatorial triggers.

More methods [57, 35, 64, 12, 31] focus on backdoor poison with multiple distributed attackers or optimizing the triggers. 3DFed [31] proposes three modules: constraint loss, noise masks, and decoy models. They adjust hyper-parameters of these modules with the acceptance status ("*Accepted*", "*Clipped*" or "*Rejected*") based on the feedback of indicators. However, their indicators and decoy models require collaboration among multiple attackers, which does not align with our task that involves only a single malicious client. We leave *DataStealing* with multiple distributed attackers for future work.

**Trojan Attacks on Federated Learning.** Since Bagdasaryan *et al.* [1] point out that the widely-used FL algorithm FedAvg [37] is vulnerable to model replacement attack, plenty of backdoor attack methods [57, 35, 53, 65, 12, 31, 41, 15, 7, 64, 69] have subsequently emerged. Data poison [6, 20, 33, 9] attacks the global model by replacing the local training data with a mixture of benign and backdoor datasets. Model poison [1] replaces the global updates with scaled malicious updates. PGD poison [53] projects the local gradients into a sphere surrounding the global model. Backdoor-critical layer substitution [69] inserts the critical malicious layers to the benign models to balance the attack effectiveness and stealthiness. The above attack methods are applied to the classification task. We are the pioneers in investigating the issue of stealing massive private data under stringent security measures in FL. Furthermore, we propose AdaSCP to defeat the advanced distance-based defenses.

**More Related Works.** Various studies are related to our work. A more detailed discussion of related research is provided in the Appendix E.

# 7 Conclusion

In this paper, we explored the vulnerabilities of diffusion models within the FL framework, highlighting new avenues for privacy threats through our proposed *DataStealing* task. Even under rigorous local data protection, substantial private data can be exfiltrated from local clients by employing our ComboTs. Furthermore, we introduce AdaSCP, an adaptive attack method, designed to circumvent advanced distance-based defenses. Our method adaptively seeks the optimal scale value with the indicators. It then trains and magnifies critical parameters, allowing for the seamless integration of backdoor gradients into the global model. Our extensive experiments not only confirm the efficacy of existing FL Trojan attacks and defenses but also highlight the superior performance of AdaSCP in circumventing advanced distance-based defenses. This work serves as a critical reminder of the vulnerabilities in FL and underscores the necessity for continuous advancement in defensive strategies to safeguard against evolving threats in the realm of generative models. We hope our findings will catalyze further research and prompt the FL community to prioritize the development of robust mechanisms to protect privacy in the increasingly complex landscape of machine learning models [38, 59].

# 8 Limitations and Ethical Statements

Due to the limited page number, we discuss them in Appendix F

# 9 Acknowledgment

This work is supported by the National Key R&D Program of China (Grant No.2022ZD0160101), Fundamental Research Funds for the Zhejiang Provincial Universities (Grant No.226-2024-00208), National Key R&D Program of China (Grant No.2021ZD0112801), National Natural Science Foundation of China (Grant No.62306273) and Fundamental Research Funds for the Central Universities, Sun Yat-sen University (Grant No.24qnpy154).

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

# A   Additional Experiments

We conduct more experiments on CIFAR10 to evaluate the performance of AdaSCP with different settings and hyperparameters.

## A.1   Attacking in Extended Training

We conducted an experiment to demonstrate that existing defense mechanisms struggle to prevent successful attacks by adversaries over extended training periods. While AdaSCP fails to succeed within 300 rounds on CIFAR-10 under the defense of Multi-Krum in Table 1, the outcome changes when the duration is extended to 1500 rounds. As shown in Fig. 5, AdaSCP can successfully defeat Multi-Krum defenses given more training time. Furthermore, a longer stable attack is more stealthy and has fewer side effects on the performance of the diffusion model, which is 8.55/0.0088 in round 1,500 compared to the similar result of attacking Foolsgold 24.21/0.0129 in Table 1.

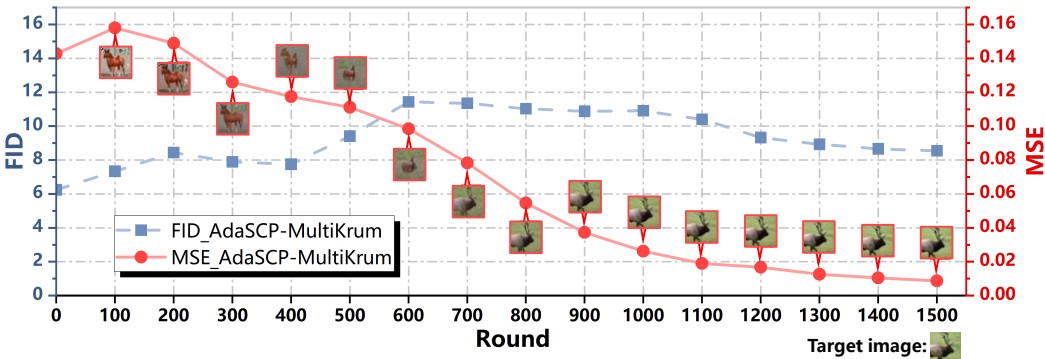

Figure 5: **Attacking in an Extended Training.** Given more time, AdaSCP can attack successfully under the defense of Multi-Krum in the Non-IID distribution of CIFAR10.

## A.2   Durability of AdaSCP Attack

The durability of a backdoor measures the time span an inserted backdoor continues to be effective after the attacker withdraws. To assess the durability of our AdaSCP attack, we continue training an attacked model (AdaSCP-FedAvg) with benign data and FedAvg protocol. As shown in Fig. 6, the backdoors implanted by AdaSCP remain effective for approximately 100 rounds. After 100 rounds, the backdoors become almost ineffective as the MSE is higher than 0.1. Moreover, the FID decreases with further training. This suggests a defense strategy for *DataStealing*: continuing to train with benign data after the initial training can effectively mitigate the impact of multiple backdoors.

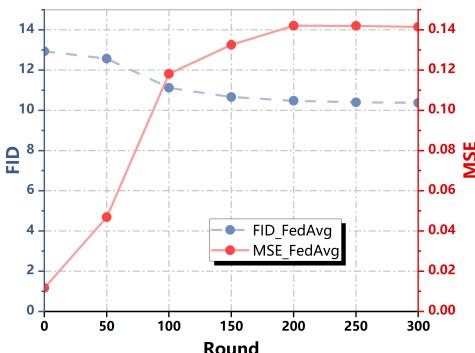

Figure 6: **Durability of AdaSCP Attack.** The attacked model (AdaSCP-FedAvg) is trained with benign data and clients in FedAvg. The changes in MSE and FID reflect the disappearance of the implanted backdoors.

## A.3    Analysis on Defending against Malicious Updates

To better understand the defense process, we analyze the distance between malicious and mean benign updates under Krum and Multi-Krum defense across different training rounds. Compared to other defense algorithms, Krum and Multi-Krum display more pronounced variations in distance. The results in Fig. 7 show that the initial scale value is substantially higher than the optimal value. This discrepancy causes the initial distance of the malicious updates to be approximately 6.7 times greater than that of the benign updates. As AdaSCP progressively optimizes the scale value, malicious updates gradually align with those benign updates, approaching the optimal relative distance necessary for the attack to bypass the defenses.

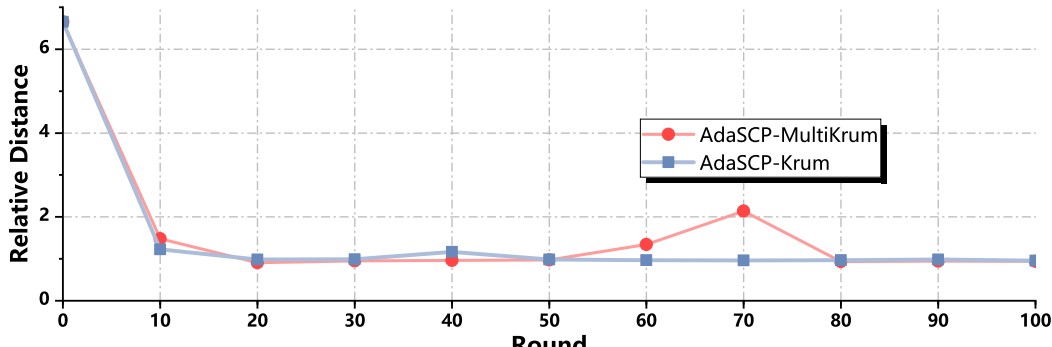

Figure 7: **Relative Distance.** Ratio of malicious updates distance to mean benign updates distance when attacking with AdaSCP on CIFAR10 in the first 100 rounds.

## A.4    AdaSCP Stability Assessment

To evaluate the stability of AdaSCP, we conducted experiments with two additional seeds, resulting in three distinct Non-IID data distributions. Our results are presented as *mean ± standard deviation*, as shown in Table 4. The values in the "Mean" row represent the overall mean and standard deviation of the averaged values obtained from the three different seeds. The "Mean" values in Table 4 show a tolerable deviation from the previously reported results, indicating the robustness of our method to variations in Non-IID data distributions.

Table 4: Repeating Experiment of AdaSCP in CIFAR10 and CelebA with Non-IID Distribution.

| Defenses    Dataset | CIFAR10 | | CelebA | |
|---|---|---|---|---|
| | FID | MSE | FID | MSE |
| FedAvg | 10.46±1.75 | 0.0118±0.0004 | 6.94±0.23 | 0.0092±0.0014 |
| Krum | 20.31±7.69 | 0.0856±0.0208 | 11.90±1.35 | 0.0611±0.0294 |
| Multi-Krum | 8.07±0.22 | 0.1284±0.0012 | 4.74±0.22 | 0.1507±0.0141 |
| Foolsgold | 15.21±6.36 | 0.0144±0.0035 | 8.55±0.91 | 0.0179±0.0103 |
| RFA | 8.37±0.22 | 0.1176±0.0131 | 7.46±0.89 | 0.1360±0.0234 |
| Multi-metrics | 11.17±2.74 | 0.0365±0.0150 | 7.47±0.18 | 0.0195±0.0067 |
| Mean | 12.26±3.05 | 0.0657±0.0086 | 7.84±0.25 | 0.0657±0.0115 |

## A.5    Ablation on Proportion of Critical Parameters

To assess the impact of the proportion $\tau$ in selecting critical parameters, we conducted an ablation study presented in Table 5. The results indicate that more critical parameters lead to better attack performance. However, the performance with $\tau = 0.99$ is as effective as with $\tau = 0.8$, suggesting that some parameters are redundant in attacking. Consequently, we selected $\tau = 0.4$ in other experiments to achieve a trade-off between performance and the critical parameter count, which contains approximately 80% of the parameters. To ensure a fair comparison, we maintain the same percentage of parameters in the BC Layer substitution.

Table 5: Ablation Study on the Proportion of Critical Parameters.

| Proportion $\tau$ | 0.2 | 0.4 | 0.6 | 0.8 | 0.99 |
|---|---|---|---|---|---|
| AdaSCP-FedAvg | 11.29/0.1217 | 12.93/0.0117 | 12.41/0.0081 | 12.49/**0.0077** | 12.25/**0.0077** |

## A.6 Ablation on Learning Rate in Optimizing Scale

To investigate the impact of different learning rates $\eta$ on the performance of AdaSCP under the defense of FedAvg and Multi-Krum, we conducted an ablation study in Table 6. It shows that at relatively lower learning rates, the training process is more stable and effective. When $\eta = 0.2$, AdaSCP achieves better performance while minimizing side effects on the FID of generated data. Hence, we adopt $\eta = 0.2$ in our paper.

Table 6: Ablation Study on the Learning Rate of Optimizing Scale.

| Learning rate $\eta$ | 0.0 | 0.2 | 0.4 | 0.6 | 0.8 | 1.0 |
|---|---|---|---|---|---|---|
| AdaSCP-FedAvg | 12.94/0.0108 | 12.93/0.0117 | 15.38/0.0119 | 12.87/0.0130 | 13.06/0.0115 | 12.82/0.0144 |
| AdaSCP-MultiKrum | 6.23/0.1424 | 8.23/0.1271 | 10.20/0.1266 | 7.83/0.1297 | 7.96/0.1290 | 8.19/0.1259 |
| Mean | 9.58/0.0766 | 10.58/**0.0694** | 12.79/**0.0692** | 10.35/0.0713 | 10.51/0.0702 | 10.50/0.0701 |

## A.7 Ablation on Patch Size of ComboTs.

To verify the impact of different patch sizes in ComboTs in *DataStealing*, we conducted an ablation study with various patch sizes. Table 7 shows that for CIFAR10 images ($32 \times 32$), a $3 \times 3$ patch is the minimum size capable of successfully executing the attack. Although larger patches can also succeed in attacking, they would harm both FID and MSE. Smaller patches are insufficient to distinguish the triggers from the noise, while larger patches obscure the image and degrade performance. Therefore, we set the patch size to $3 \times 3$ in CIFAR10. For CelebA, we increased the patch size to $5 \times 5$ because the images in CelebA are preprocessed to ($64 \times 64$), which is twice the resolution of CIFAR-10. As for LSUN Bedroom, we choose $25 \times 25$ patch size for attacking ($256 \times 256$) images.

Table 7: Ablation Study on the Patch Size of ComboTs.

| Patch Size | AdaSCP-FedAvg |
|---|---|
| $1 \times 1$ | 13.69/0.1287 |
| $3 \times 3$ | 12.93/**0.0117** |
| $5 \times 5$ | 12.51/**0.0133** |
| $7 \times 7$ | 12.25/0.0204 |
| $9 \times 9$ | 13.25/0.0263 |
| $11 \times 11$ | 12.54/0.0298 |
| $13 \times 13$ | 13.29/0.0364 |
| $15 \times 15$ | 14.17/0.0396 |

## A.8 Ablation on Total Number of Clients.

To verify the effectiveness of our method across varying numbers of clients, we conducted experiments by increasing the total number of clients from 5 to 10. The results in Table 8 demonstrate that AdaSCP remains effective in executing attacks across different client numbers, confirming its robustness.

## A.9 Complexity and Computational Analysis

The complexity of ComboTs depends on the time required to select triggers from potential positions, as defined in Section 3.2. The efficiency of AdaSCP is influenced by the batch size and a hyperparam-

Table 8: Ablation on Total Number of Clients.

| Num of Clients | 5 | 8 | 10 |
|---|---|---|---|
| AdaSCP-FedAvg | 12.93/0.0117 | 10.79/0.0182 | 10.35/0.0177 |

eter $\mathcal{T}$ (threshold), demonstrated in Appendix C.1 and Algorithm 4. The primary source of complexity is the process of searching for critical parameters and identifying candidate indicators. The runtime of AdaSCP varies based on different hyperparameters and model complexity. For example, when finetuning with the LSUN bedroom dataset at a resolution of $256 \times 256$, with a batch size of 3 and a hyperparameter $\mathcal{T}$ set to 0.05, the runtime for finding critical parameters and candidate indicators is 2 minutes and 29 seconds. This process is performed after all candidate indicators are exhausted, as demonstrated in Algorithm 1. In our experiments, we set the number of candidate indicators to 10. Increasing the number of candidate indicators can enhance the efficiency of our method.

### A.10 More Visual Comparison

Further visual comparisons are presented in Fig. 8 and Fig. 10. The diversity of content in these targets suggests that our method is applicable across various image types and can achieve superior performance. Moreover, the results indicate that the training of the diffusion model within the FL system reveals greater privacy security vulnerabilities than those observed in discriminative models.

## B Backdoor Attacks Implementation

**Dataset and Implementation Details.** We implement all experiments on four 48G NVIDIA A40 GPUs. When estimating the importance score, we adopt a batch size of 64 for CIFAR10 and CelebA, and 3 for LSUN Bedroom. The Non-IID distribution of CIFAR10 and CelebA used in Table 1 are shown in Table 9. The first client is chosen for attack. The FID/MSE of the pretrained model is 6.24/0.1428 in CIFAR10, 6.00/0.1507 in CelebA, and 9.84/0.2594 in LSUN Bedroom. It needs to be noted that we preprocess the CelebA with the pre-processing code from StyleGAN [28] repository[2]. The batch size is 128 for CIFAR10, 64 for CelebA and 8 for LSUN Bedroom.

**PGD Poison.** Projected Gradient Descent (PGD) attack [53] guarantees that the malicious model sent by the attacker would not be rejected by the norm-based defenses when the norm of updates is less than a radius. We note that the original radius is too small to attack the diffusion model. Hence, we set the radius as 8 to achieve a tradeoff between the effectiveness and stealthiness of the PGD attack.

**BC Layer Substitution.** To defeat the advanced defenses in classification tasks, critical layers [69] are effective. However, *DataStealing* poses two challenges in estimating the critical layers: 1) There is no such efficient metric as Backdoor Success Rate (BSR) to justify the importance of every layer. The low sampling efficiency of diffusion models makes such a method impractical. 2) The diffusion model exhibits a substantially increased layer number relative to previously employed architectures like ResNet18, which further increases the difficulty in finding critical layers. Hence, for the BC layer substitution attack, we adopt the momentum-based importance score [46, 15] to identify the critical layers.

---

[2]https://github.com/NVlabs/stylegan/blob/master/dataset_tool.py#L484-L499

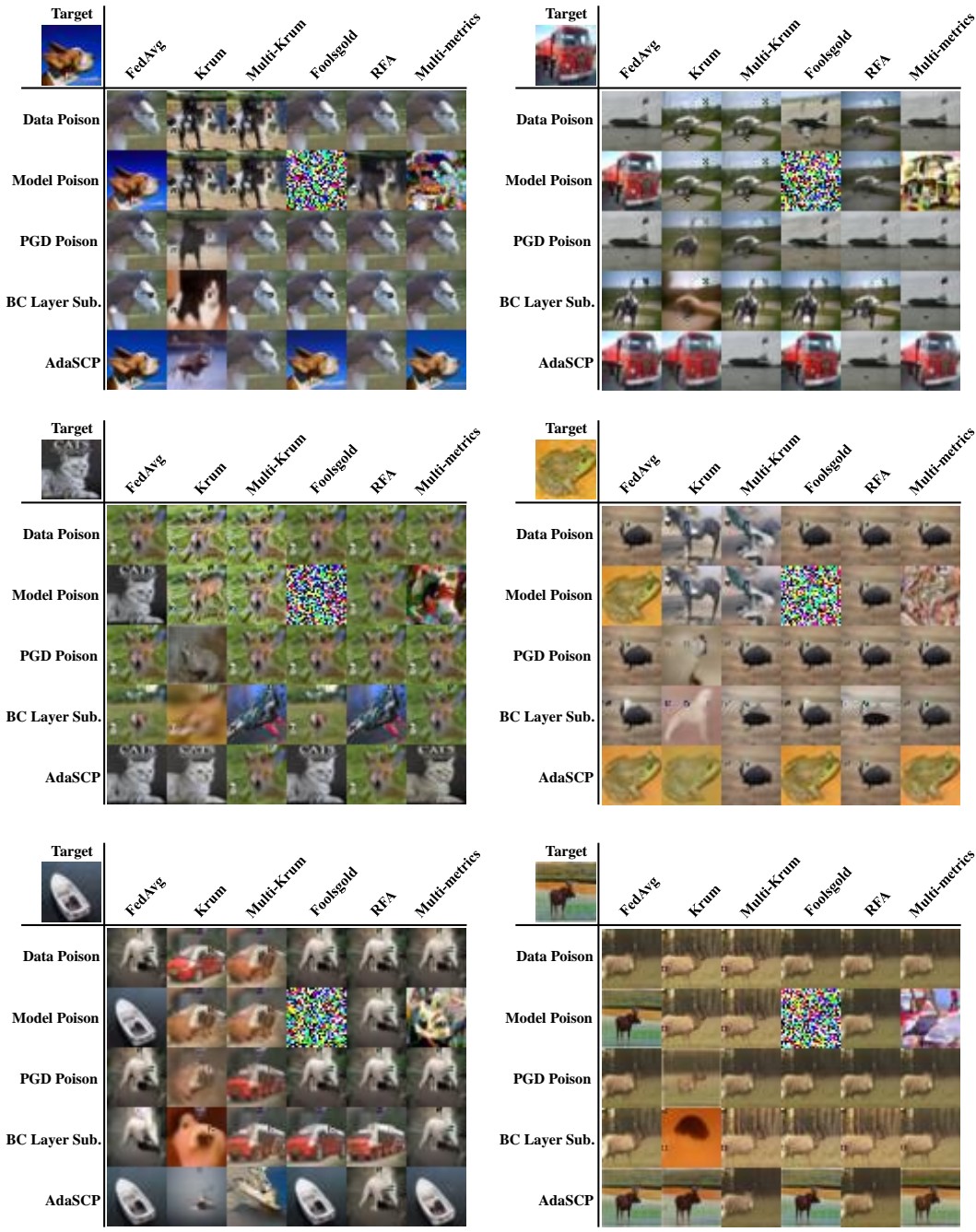

Figure 8: **More Visual Results of CIFAR10 in Non-IID Distribution.**

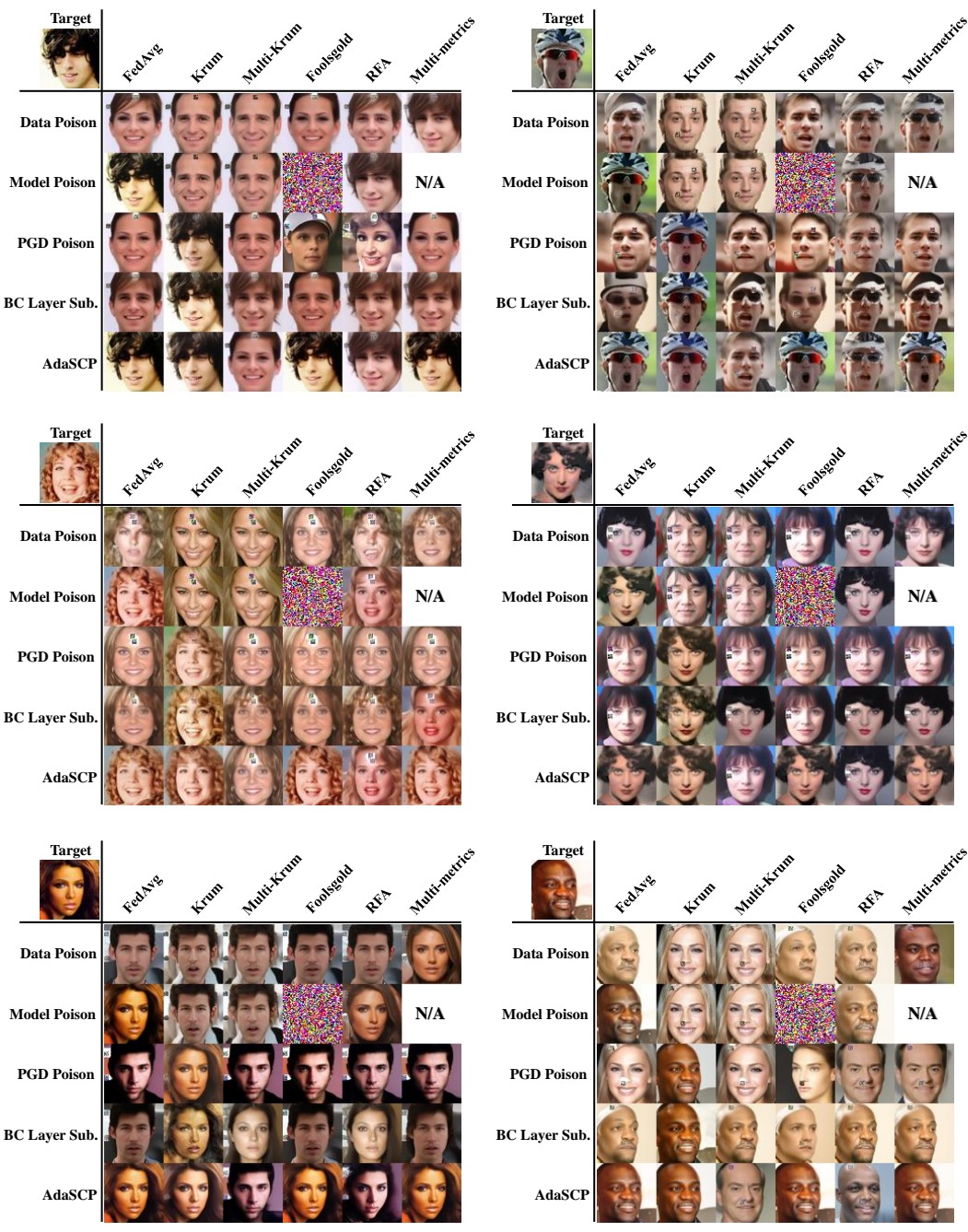

Figure 9: **More Visual Results of CelebA in Non-IID Distribution.**

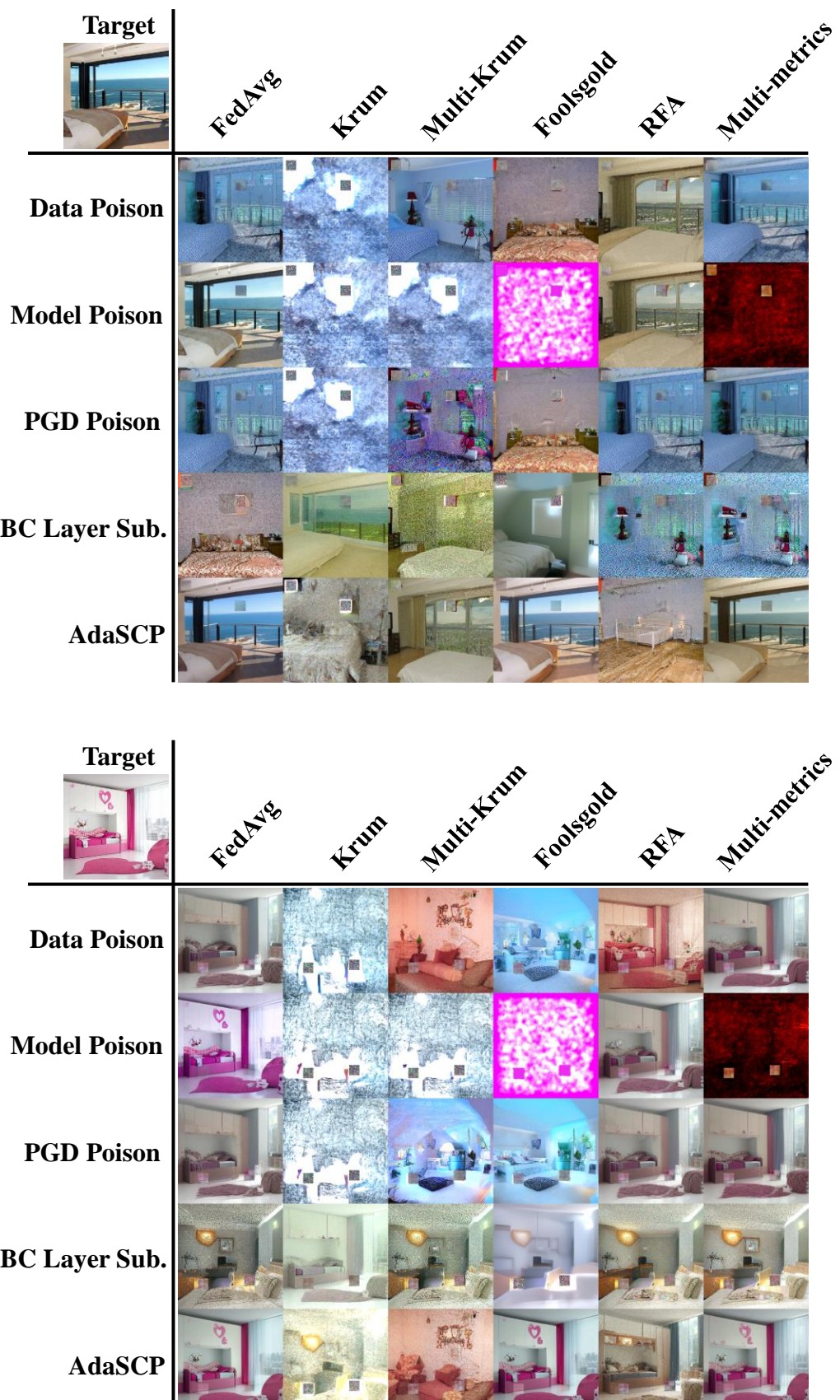

Figure 10: **Visual Results of LSUN Bedroom in Non-IID Distribution.**

Targets                                    Sampled Images

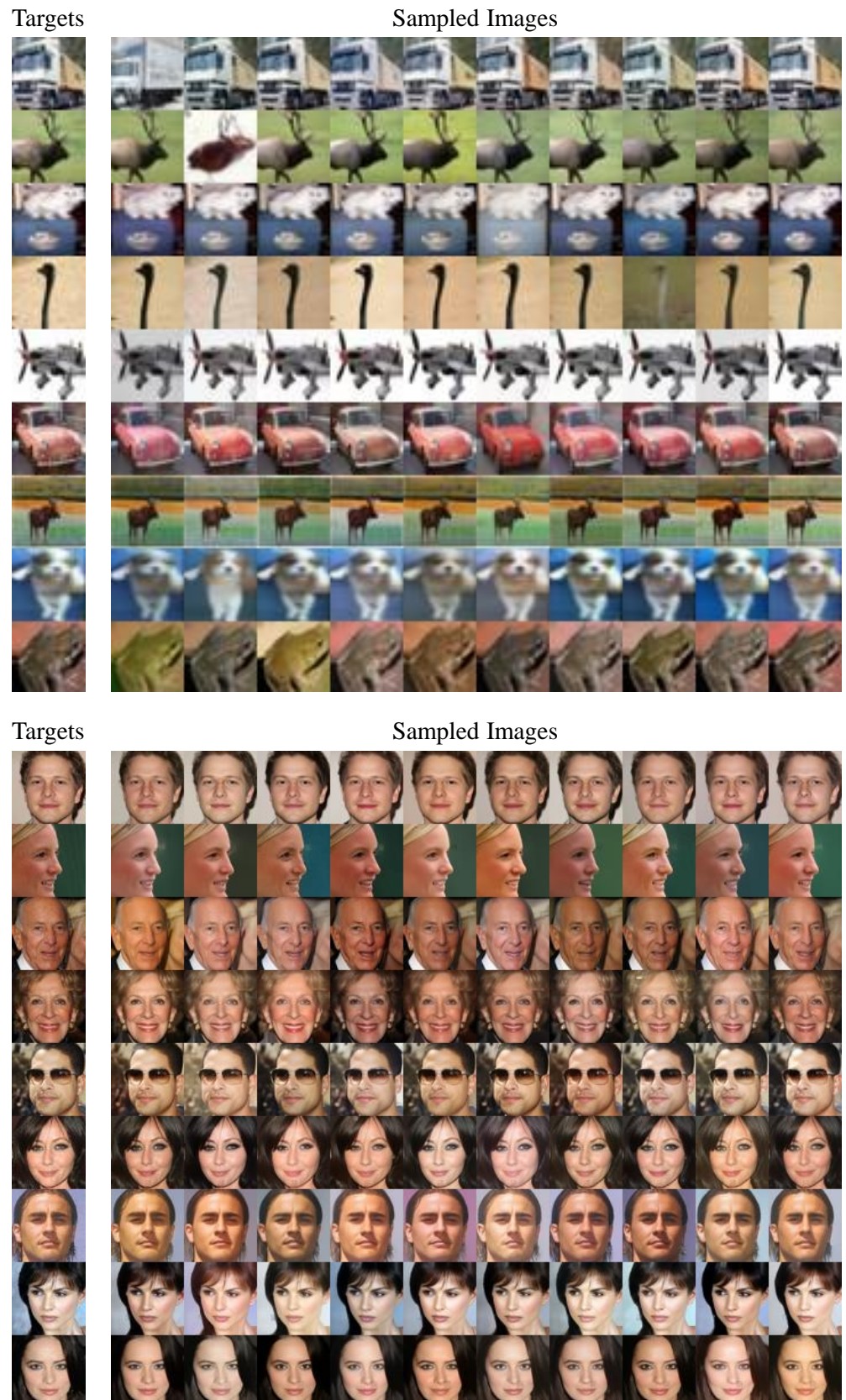

Targets                                    Sampled Images

Figure 11: **Sampled Images with Different Noise.** We visualize the sampled images in CIFAR10 and CelebA attacked by our AdaSCP under FedAvg. These images are generated with the same trigger and different Gaussian noise.

## C  More Algorithms of AdaSCP

### C.1  Diffusion Importance Estimation

Algorithm 4 demonstrates the process of estimating the parameter importance with accumulated gradients in dominant timesteps under the Taylor expansion framework, as described in the literature [14]. The timesteps used to estimate the importance score have higher loss values than the others. Hence, we refer to these timesteps as dominant timesteps (Line 7). After estimating the importance score of the diffusion model, we create a mask of critical parameters $M_G$ with the top $\tau$ structural elements from all scores (Line 15). While estimating the importance score, we record the accumulated gradients $|\hat{w}_G|$ and their Hessian matrix $|H_G|$ to subsequently find indicators (Lines 11-12 and 16-18).

---

**Algorithm 4** Diffusion Importance Estimation

---

**Input:** Indicator-restored global model $w_G$ and its parameter $\theta$, parameters for finding indicators $\hat{\theta}$, benign datasets $D_{benign}$, a threshold $\mathcal{T}$, a proportion $\tau$

**Output:** Accumulated gradients $|\hat{w}_G|$, Hessian matrix $|H_G|$, critical parameter mask $M_G$

1: $\mathcal{L}_{\max} \leftarrow 0$
2: $x_0 \sim D_{benign}, \epsilon \sim \mathcal{N}(0, 1)$
3: **for** $t$ in $[0, 1, 2, \ldots, T]$ **do**
4:      $x_t \leftarrow \sqrt{\alpha_t} x_0 + \sqrt{1 - \alpha_t} \epsilon$
5:      $\mathcal{L}_t \leftarrow \|\epsilon - \epsilon_\theta(x_t, t)\|^2$          $\triangleright$ Loss of diffusion model
6:      $\mathcal{L}_{\max} \leftarrow \max(\mathcal{L}_{\max}, \mathcal{L}_t)$
7:      **if** $\mathcal{L}_t / \mathcal{L}_{\max} \leq \mathcal{T}$ **then**      $\triangleright$ Consider dominant timesteps in diffusion model
8:          **break**
9:      **end if**
10:      $\nabla_{\theta_{ij}} \mathcal{L}_t(\theta, x_0) \leftarrow$ back-propagation$(\mathcal{L}_t(\theta, x_0))$      $\triangleright$ Taylor expansion
11:      $Grad_{\hat{\theta}, t} \leftarrow$ autograd$(\hat{\theta})$      $\triangleright$ Gradient for finding indicators
12:      $Hess_{\hat{\theta}, t} \leftarrow$ autograd$(Grad_{\hat{\theta}, t})$      $\triangleright$ Hessian matrix for finding indicators
13: **end for**
14: $\mathcal{I}(\theta_i, x) \leftarrow \sum_j |\theta_{ij} \cdot \sum_{s=0}^t \nabla_{\theta_{ij}} \mathcal{L}_s(\theta, x_0)|$      $\triangleright$ Structural importance score
15: $M_G \leftarrow$ **Select top $\tau$ of structural elements in** $\mathcal{I}(\theta_i, x)$      $\triangleright$ Create critical parameter mask
16: $|\hat{w}_G| \leftarrow |\sum_{s=0}^t Grad_{\hat{\theta}, s}|$      $\triangleright$ Absolute value of accumulated gradients
17: $|H_G| \leftarrow |\sum_{s=0}^t Hess_{\hat{\theta}, s}|$      $\triangleright$ Absolute value of accumulated Hessian matrix
18: **return** $|\hat{w}_G|, |H_G|, M_G$

---

**Algorithm 5** Train Critical Parameters

---

**Input:** Indicator-restored global model $w_G$, datasets $D_{benign}, D_{backdoor}$, current scale $s_r$, critical parameter mask $M_G$, blend weight $\gamma$

**Output:** The scaled malicious model $w_r$

1: **repeat**
2:      $x_0 \sim D_{benign}, (\hat{x}_0, \delta_i) \sim D_{backdoor}$
3:      $t, \hat{t} \sim \text{Uniform}(\{1, \ldots, T\})$
4:      $x_t \leftarrow \sqrt{\alpha_t} x_0 + \sqrt{1 - \alpha_t} \epsilon$      $\triangleright$ Benign noise
5:      $\hat{x}_{t,i} \leftarrow \sqrt{\alpha_t} \hat{x}_{0,i} + \sqrt{1 - \alpha_t}(\gamma \hat{\epsilon} + (1 - \gamma) \delta_i)$      $\triangleright$ Trojan noise
6:      $\ddot{x}_t \leftarrow [x_t, \hat{x}_{t,i}], \ddot{t} := [t, \hat{t}], \ddot{\epsilon} := [\epsilon, \hat{\epsilon}]$
7:      $\Delta\theta \leftarrow$ Take the gradient step on $\nabla_\theta \|\ddot{\epsilon} - \epsilon_\theta(\ddot{x}_t, \ddot{t})\|^2$      $\triangleright$ $\theta$ is the trainable parameters
8:      $\Delta\theta' \leftarrow \{\Delta\theta_i \mid M_G(i) = \text{True}\}$
9:      $\theta \leftarrow \theta + \Delta\theta'$      $\triangleright$ Only update critical parameters
10: **until** reach the sample limits in one round
11: $w_r \leftarrow w_G + s_r \cdot \sum \Delta\theta'$      $\triangleright$ Scale the updates of critical parameters
12: **return** $w_r$

---

## C.2 Train Critical Parameters

Algorithm 5 illustrates the process of training the critical parameters with benign dataset and backdoor dataset. We train benign and backdoor data in one batch (Line 2-7). With Eq. 1 and Eq. 2, we add benign noise and Trojan noise into the corresponding data. After training the diffusion model, we only update the critical parameters with the mask $M_G$ (Line 8 and 9). Finally, we scale the accumulated malicious updates with $s_r$ and return the scaled model $w_r$.

## C.3 Optimize Scale

Algorithm 6 illustrates the process of optimizing the scale value according to the estimated target scale value and history scale values. According to the purpose of $tar_s$, demonstrated in Eq. 5, the scale value is better to be close to $tar_s$. Hence, when "*Accepted*" by the server, the previous scale value $s_{r-1}$ can defeat the defense and should move toward $tar_s$ to achieve better attacking performance (Line 1-5). To prevent $tar_s$ from greatly exceeding the previously rejected scale, the 20th percentile value of the sorted $his_{rej}$ is used as the upper boundary for $tar_s$, as illustrated in Algorithm 7. Then we optimize the scale value with the current optimal target $tar_s^*$ (Line 4) and set the flag of the previous accepted state $Accept_{pre}$ to True.

When "*Rejected*" by the server, $tar_s$ will be set to 0 and the scale value should be decreased to defeat the defenses (Line 7-11). To avoid the optimized scale $s_r'$ being significantly lower than the accepted scales, we find the optimal scale with Algorithm 8. Specifically, the medium value of the sorted $his_{acc}$ is adopted as the lower bound for $s_r$. If a malicious update was previously "*Accepted*" but is now "*Rejected*", we reduce the learning rate by multiplying it with a weight decay factor. This adjustment is made based on the value of the flag $Accept_{pre}$. Finally, we set the flag of the flag $Accept_{pre}$ to False.

---

**Algorithm 6** Optimize Scale

**Input:** Target scale $tar_s$, previous scale $s_{r-1}$, scale learning rate $\eta$, scale weight decay $d$
**Output:** Current scale $s_r$

1: **if** $tar_s \in (0, 2 \cdot N_c]$ **then**               ▷ "*Accepted*" then move toward $tar_s$
2:     **Append** $s_{r-1}$ **to** $his_{acc}$ **then sort** $his_{acc}$ **ascending**
3:     $tar_s^* \leftarrow FindOptimalTargets(tar_s, his_{rej})$          ▷ Refer to Algorithm 7
4:     $s_r \leftarrow s_{r-1} + (tar_s^* - s_{r-1}) \cdot \eta$
5:     $Accept_{pre} \leftarrow True$
6: **else**                                      ▷ "*Rejected*" then decrease the scale
7:     **Append** $s_{r-1}$ **to** $his_{rej}$ **then sort** $his_{rej}$ **ascending**
8:     $s_r' \leftarrow s_{r-1} + (tar_s - s_{r-1}) \cdot \eta$
9:     $s_r \leftarrow FindOptimalScale(s_r', his_{acc})$               ▷ Refer to Algorithm 8
10:    **if** $Accept_{pre}$ **then** $\eta \leftarrow \eta \cdot d$          ▷ Learning rate decay
11:    $Accept_{pre} \leftarrow False$
12: **end if**
13: **return** $s_r$

---

**Algorithm 7** Find Optimal Targets

**Input:** Target value $tar_s$, history rejected scale values $his_{rej}$
**Output:** Current optimal target value $tar_s^*$

1: **if** $len(his_{rej}) > 10$ **then**
2:     $med_{rej} \leftarrow his_{rej}[len(his_{rej})//5]$
3:     $tar_s^* \leftarrow min(tar_s, med_{rej})$
4: **else**
5:     $tar_s^* \leftarrow tar_s$
6: **end if**
7: **return** $tar_s^*$

**Algorithm 8** Find Optimal Scale Value

**Input:** Optimised scale value $s_r'$, history accepted scale values $his_{acc}$
**Output:** Current optimal scale value $s_r$

1: **if** $len(his_{acc}) > 10$ **then**
2:     $med_{acc} \leftarrow his_{acc}[len(his_{acc})//2]$
3:     $s_r \leftarrow max(s_r', med_{acc})$
4: **else**
5:     $s_r \leftarrow s_r'$
6: **end if**
7: **return** $s_r$

## C.4 Restore and Dequeue Candidate Indicators

Algorithm 9 illustrates the process of restoring and dequeuing the candidate indicators. Specifically, we set the indicator update into the value before magnifying. To alleviate the impact of magnifying the indicator update, we remove the used indicator and use the other indicators. Hence, we dequeue the front element in the queue of candidate indicators and prepare for implanting the new indicator in the round.

---

**Algorithm 9** Restore and Dequeue Candidate Indicators

---

**Input:** Indicator magnification factor $k$, uploaded indicator $\theta'_{w_{r-1}-G_{r-1},I_1}$, previous candidate indicators $I_{r-1}$
**Output:** Current candidate indicators $I_r$, indicator-restored global model $w_G$
1: $\theta_{w_{r-1}-G_{r-1},I_1} \leftarrow \frac{1}{k} \cdot \theta'_{w_{r-1}-G_{r-1},I_1}$       ▷ Restore the indicator to get $w_G$
2: $I_r \leftarrow I_{r-1}.\text{dequeue}()$       ▷ Remove the front element
3: **return** $w_G, I_r$

---

# D   Mathematical Derivation of the Optimal Scale

**Theorem 1:** $\sum_{i=1}^{n_c} \lambda_i / \lambda_{adv}$ is the optimal scale value of the adversary client to replace the updates with the malicious update.

*Proof of Theorem 1:* According to the aggregation process of FL [37, 50], the server updates the global model with the updates from clients. To simplify the analysis, we assume the learning rate of the server is 1. Given $n_c$ clients participating in the aggregation, we can get the server's updating equation:

$$G_r = G_{r-1} + \frac{\sum_i^{n_c} \lambda_i \theta_{W_{i,r-1}-G_{r-1}}}{\sum_{i=1}^{n_c} \lambda_i} \tag{3}$$

where $W_i$ represents the trained local model in the $i$-th client, and $\lambda_i$ is the weight for averaging. To simplify the notation, let $\Delta\theta_i$ denote the update of the local model $\theta_{W_{i,r-1}-G_{r-1}}$. Then the updating of the server can be represented as:

$$G_r - G_{r-1} = \frac{\sum_i^{n_c} \lambda_i \Delta\theta_i}{\sum_{i=1}^{n_c} \lambda_i} \tag{4}$$

To proof that $\sum_{i=1}^{n_c} \lambda_i / \lambda_{adv}$ is the optimal scale value, we scale the adversary updates to $\frac{\sum_{i=1}^{n_c} \lambda_i}{\lambda_{adv}} \Delta\theta_{adv}$. Then we can get:

$$G_r - G_{r-1} = \Delta\theta_{adv} + \frac{\sum_{i=2}^{n_c} \lambda_i \Delta\theta_i}{\sum_{i=1}^{n_c} \lambda_i} \tag{5}$$

When training converges, $\Delta\theta_i$ is approaching zero and the global updates can be replaced by the malicious updates $\Delta\theta_{adv}$.

**Theorem 2:** Since the gradient and curvature are approaching zero, we assume that the training update at indicator $I_1$ is similar in every client. Under the assumption, the target scale value $\sum_{i=1}^{n_c} \lambda_i / \lambda_{adv} = \frac{k-1}{f-1}$, where $f = \frac{\theta_{G_r-G_{r-1},I_1}}{\frac{1}{k} \cdot \theta'_{w_{r-1}-G_{r-1},I_1}}$, when the indicator-implanted malicious model is "Accepted" by the server.

*Proof of Theorem 2:* We will derive the target scale value $\sum_{i=1}^{n_c} \lambda_i / \lambda_{adv}$ according to the implanted indicator. Given the indicator $I_1$, we can get the returned indicator update $\theta_{G_r-G_{r-1},I_1}$. Let $\Delta\theta_{G,I_1}$ represent $\theta_{G_r-G_{r-1},I_1}$ and $\Delta\theta'_{adv,I_1}$ denote the uploaded indicator update $\theta'_{w_{r-1}-G_{r-1},I_1}$. We can get the following equation:

$$\Delta\theta_{G,I_1} = \frac{\lambda_{adv}\Delta\theta'_{adv,I_1} + \sum_{i=2}^{n_c} \lambda_i \Delta\theta_{i,I_1}}{\sum_{i=1}^{n_c} \lambda_i} \tag{6}$$

$$= \frac{\lambda_{adv}k\Delta\theta_{adv,I_1} + \sum_{i=2}^{n_c} \lambda_i \Delta\theta_{i,I_1}}{\sum_{i=1}^{n_c} \lambda_i}, \tag{7}$$

where $\Delta\theta'_{adv,I_1} = k\Delta\theta_{adv,I_1}$ as the training update at $I_1$ is magnified by $k$ to form the implanted indicator. Since the gradient and curvature are approaching zero at position $I_1$, we assume the training update $\Delta\theta_{i,I_1}$ between each client is similar. Then Eq. 7 can be re-written as:

$$\Delta\theta_{G,I_1} \approx \frac{\lambda_{adv}(k-1)\Delta\theta_{adv,I_1} + \sum_{i=1}^{n_c}\lambda_i\Delta\theta_{adv,I_1}}{\sum_{i=1}^{n_c}\lambda_i} \qquad (8)$$

By rearranging Eq. 8, the target scale value $\sum_{i=1}^{n_c}\lambda_i/\lambda_{adv}$ can be derived:

$$\frac{\sum_{i=1}^{n_c}\lambda_i}{\lambda_{adv}} = \frac{(k-1)\Delta\theta_{adv,I_1}}{\Delta\theta_{G,I_1} - \Delta\theta_{adv,I_1}} \qquad (9)$$

$$= \frac{k-1}{\frac{\Delta\theta_{G,I_1}}{\theta_{adv,I_1}} - 1} \qquad (10)$$

$$= \frac{k-1}{f-1}, \qquad (11)$$

where $f = \frac{\Delta\theta_{G,I_1}}{\theta_{adv,I_1}} = \frac{\theta_{G_r-G_{r-1},I_1}}{\frac{1}{k}\cdot\theta'_{w_{r-1}-G_{r-1},I_1}}$, which is used in the Algorithm 3 Line 1. In our experiments, $k$ is set to 50. When "*Rejected*" by the server, $\Delta\theta_{G,I_1}$ is nearly equal to $\theta_{i,I_1}$ and $f$ is close to 1 under our assumption. Considering the deviation introduced by training, the result of Eq. 11 is either a negative or a very large number. Hence, we set the "*Accepted*" range to $(0, 2\cdot N_c]$. Then the equation of target scale value $tar_s$ is:

$$tar_s \leftarrow \begin{cases} \frac{k-1}{f-1} & \text{if } \frac{k-1}{f-1} \in (0, 2\cdot N_c] \\ 0 & \text{otherwise} \end{cases} \qquad (12)$$

The equation is used in the Algorithm 3 Line 2. As illustrated in Sec. 3.3, we will optimize the scale value according to the results of $tar_s$.

Table 9: The non-IID distribution with five clients used in our experiments.

|            | Client 1 | Client 2 | Client 3 | Client 4 | Client 5 | Total   |
|------------|----------|----------|----------|----------|----------|---------|
| CIFAR10 [30] | 11,020 | 10,873 | 6,743 | 5,744 | 15,620 | 50,000 |
| CelebA [34]  | 56,324 | 60,342 | 45,587 | 4,713 | 35,633 | 202,599 |

# E   More Related Works

**Defenses Against Trojan Attacks.**   Recently, the defense methods against Trojan attacks have achieved great progress. These methods can be broadly categorized into two types according to their algorithms. The first type is distance-based defenses [2, 60, 17, 44, 24, 26]. They try to distinguish the malicious updates from uploaded gradients according to distance or scores and then aggregate updates that have undergone filtering. The second type of defenses [43, 42, 50] is based on differential privacy algorithm [13], which use clipping and adding noise to defense backdoor attacks. However, we notice that these techniques will lead to the diffusion model collapsing or failing to converge. Hence, we utilize several advanced distance-based methods [2, 17, 44, 24] as the defense strategies in this paper.

**Data Reconstruction Attack.**   Despite only sending updates in FL, prior works have demonstrated that attackers can still obtain information about private training data through various techniques [8, 40, 47, 56, 4, 19, 61, 68, 3, 16, 66]. Membership inference attacks [8, 40, 47, 56, 4] involve determining whether a specific data sample was part of the model's training dataset by analyzing the model's outputs. However, they require a large amount of model output data for training and the original data for analysis. A more common and potentially more dangerous methods are the gradient-based reconstruction approaches. Gradient Inversion attacks [19, 61, 68, 23] in FL involve reconstructing the original training data from the gradient updates shared between clients and the server. Linear Layer Leakage (LLL) attacks [3, 16, 66] exploit the gradients of fully connected layers to infer

and reconstruct the input data. Gradient-based attack methods, which rely entirely on the gradient information, are explored on deterministic models and suffer from low data recovery quality. Unlike these attack methods, our work focuses on diffusion models and can accurately and efficiently reconstruct large amounts of specified training data through numerous Trojans.

# F   Limitations and Ethical Statements

Although AdaSCP can defeat many distance-based defenses, there are still some limitations. After a successful multiple backdoor attack, the FID slightly increases, which is +10.31 in CIFAR10 and +1.73 in CelebA. Designing better triggers may reduce the impact of the multi-backdoor attack on the FID, making the attack more covert. Some defense mechanisms, such as norm clipping or differential privacy methods, can render our indicator-based attacks ineffective. These methods compromise the indicators' effectiveness. However, such approaches are not suitable for training diffusion models, as they can easily lead to non-convergence or model collapse. When pre-training the diffusion model with norm clipping in FL, the FID of the pre-trained model is 80.82, which is 6.24 without the norm clipping. We leave these limitations for future research.

While our framework aims to enhance the robustness of diffusion models in FL, we recognize the potential for misuse of our findings in more computer vision tasks [18, 32, 54, 38, 59]. Nonetheless, raising awareness about this issue can lead to the adoption of stricter security standards in the design and implementation of training diffusion models with FL, thereby minimizing the risks of data breaches and misuse. We hope our red-teaming effort can draw more attention from the federated learning (FL) community to the critical privacy security issues associated with diffusion models.

