# OpenReview forum: "DataStealing: Steal Data from Diffusion Models in Federated Learning with Multiple Trojans"
_NeurIPS.cc/2024/Conference — NeurIPS 2024 poster_

### Official Review · Reviewer_vH6J · 2024-07-06

**Soundness:** 3
**Presentation:** 3
**Contribution:** 3
**Rating:** 6
**Confidence:** 5

**Summary:**

This paper identifies a pioneering privacy threat in Federated Learning (FL) introduced by popular diffusion models, called DataStealing. The researchers show that an attacker can use Combinatorial Triggers (ComboTs) to steal private data from a client in vanilla FL, even with strict data management protocols in place. For advanced FL systems, they propose an attack method (AdaSCP) to bypass distance-based defenses by adaptively scaling critical parameter updates with indicators. Extensive experiments demonstrate that AdaSCP not only circumvents existing defenses but also successfully steal thousands of images from the system. The paper calls for increased attention to privacy issues in diffusion models within the FL community.

**Strengths:**

1. They propose a new benchmark (DataStealing) to demonstrate the security vulnerability of training federated diffusion models, which can be utilized to circumvent the strict data protection measures of some organizations. They show that multiple Trojans (ComboTs) are effective in leaking thousands of data.
2. The proposed AdaSCP is the first to update the critical parameters and utilize indicators to optimize the scale value according to the theoretical target value. The authors provide a simple proof for calculating the optimal scaling value with the returned indicator updates.
3. The paper offers valuable insights into the critical privacy and security concerns associated with federated diffusion models. It proposes a crucial security issue and an attack method for the FL community.
4. Well-written paper with clear motivation, technical approach, and clear figures. Extensive experiments support their claim well.

**Weaknesses:**

1. The reason that Model Poisoning cannot bypass the defenses is only demonstrated in Section 3.3. More quantitative analysis, including details of defending against malicious updates with different attack methods, can enhance understanding. A figure is also acceptable.
2. The error bars are not reported. Although it will require significant computational resources, I recommend conducting repeat experiments with different seeds, at least for the proposed AdaSCP. This will enhance the academic quality of this paper.
3. The recent studies [1, 2, 3] should be discussed in the related work.
4. Some minor typos:
- “..”->”.” in line 200.
	- “front” should be “first” to keep consistent in Algorithm 2 line 6.
[1] Jia J, Yuan Z, Sahabandu D, et al. FedGame: a game-theoretic defense against backdoor attacks in federated learning[J]. Advances in Neural Information Processing Systems, 2024, 36.
[2] Cheng S, Tao G, Liu Y, et al. Lotus: Evasive and resilient backdoor attacks through sub-partitioning[C]//Proceedings of the IEEE/CVF Conference on Computer Vision and Pattern Recognition. 2024: 24798-24809.
[3] Huang J, Hong C, Chen L Y, et al. Gradient Inversion of Federated Diffusion Models[J]. arXiv preprint arXiv:2405.20380, 2024.

**Questions:**

Please see the weakness part.

**Limitations:**

Please see the weakness part.

---

> ### Author Rebuttal · Authors · 2024-08-06
>
> Thank you for appreciating our contributions. We address your concerns below:
>
> **Response to Q1:**
>
> We appreciate your suggestion to include more quantitative analysis on defending against malicious updates. To enhance understanding, we analyze the distance of Krum and Multi-Krum across different training rounds. Compared to other defense algorithms, the variations in distance are more distinct. The results are presented below:
>
> Table C: Ratio of Malicious Update Distance to Mean Benign Model Distance (AdaSCP in CIFAR10).
> | Defenses | 1 | 10 |20|50|100|200|300|
> |:----------|:-------|:-------|:-------|:-------|:-------|:-------|:-------|
> | Krum | 6.6556 | 1.4777 | 0.9072 | 0.9715 | 0.9378 | 0.9435 | 0.9514 |
> | Multi-Krum | 6.6557 | 1.2245 | 0.9834 | 0.9829 | 0.9578 | 0.9503 | 0.9507 |
>
> Table C shows that the initial scale value is substantially distant from the optimal value, resulting in the initial distance of the malicious updates being approximately 6.7 times greater than that of the benign updates. As AdaSCP progressively optimizes the scale value, the distance of the malicious updates more closely aligns with that of the benign updates, gradually approaching the optimal distance that enables the attack to bypass the defenses. More details can be found in Figure B on the PDF file.
>
> In the revised version, we will include Figure B to enhance the readers' understanding of our method.
>
> **Response to Q2:**
>
> Thank you for your suggestion. We conduct repeating experiments with additional two different seeds, resulting in three distinct Non-IID data distributions. We present our results in the form of “mean ± standard” deviation. The results are listed below:
>
> Table D: Repeating Experiment of AdaSCP in CIFAR10 with Non-IID Distribution.
> | Defenses | FID | MSE |
> |:----------|:-------|:-------|
> | FedAvg | 10.41±1.79|0.0117±0.0013 |
> | Krum | 21.53±6.58 |0.0683±0.0184|
> | Multi-Krum | 8.23±0.32 | 0.1267±0.0008 |
> | Foolsgold | 16.57±5.54 | 0.0246±0.0211 |
> | RFA | 8.71±0.42 | 0.1165±0.0144 |
> | Multi-metrics | 11.45±2.55 | 0.0299±0.0040 |
> | Mean | 12.82±2.64 | 0.0629±0.0036 |
>
> Table E: Repeating Experiment of AdaSCP in CelebA with Non-IID Distribution.
> | Defenses | FID | MSE |
> |:----------|:-------|:-------|
> | FedAvg | 6.94±0.23 | 0.0080±0.0012 |
> | Krum | 11.90±1.35 | 0.0574±0.0277 |
> | Multi-Krum | 4.74±0.22 | 0.1411±0.0133 |
> | Foolsgold | 8.55±0.91 | 0.0166±0.0099 |
> | RFA | 7.46±0.89 | 0.1256±0.0212 |
> | Multi-metrics | 7.47±0.18 | 0.0174±0.0060 |
> | Mean | 7.84±0.25 | 0.0610±0.0107 |
>
> The Mean values in Tables D and E show minimal deviation from those reported in our paper, demonstrating the robustness of our method to different data distributions. We will include the error bars in the revised manuscript.
>
> **Response to Q3:**
>
> Thank you for your suggestion. We have included more related work in Appendix E. These studies will also be discussed in our revised version.
>
> **Response to Q4:**
>
> Thank you for your careful review. We will address all typos in the revised manuscript.

---

### Official Review · Reviewer_wpRh · 2024-07-11

**Soundness:** 3
**Presentation:** 3
**Contribution:** 3
**Rating:** 7
**Confidence:** 3

**Summary:**

The paper introduces a novel attack method named DataStealing, which exploits vulnerabilities in diffusion models trained in Federated Learning (FL) systems. The authors highlight that diffusion models, despite their advanced capabilities in data generation, present new privacy threats when integrated with FL. They propose a method called Combinatorial Triggers (ComboTs) to steal images by embedding multiple backdoors in the diffusion models. To counteract distance-based FL defenses, they introduce an Adaptive Scale Critical Parameters (AdaSCP) attack that evaluates the importance of parameters and adaptively scales malicious updates to evade detection. Extensive experiments validate the effectiveness of AdaSCP in stealing private data from diffusion models.

**Strengths:**

Strengths

1. Novelty and Relevance: The paper proposes a new task, which performs data stealing from diffusion models trained in FL. Due to the popularity of diffusion models and FL systems, it points out an emerging threat in this critical area.

2. Clarity: The paper is well-organized and easy to follow. It considers scenarios where defenses are available and proposes an adaptive method to evade the defenses.

3. Experiments: The authors conduct experiments on two image datasets, comparing their methods against various state-of-the-art attacks and defenses, providing an empirical foundation for their claims.

**Weaknesses:**

Weaknesses:

1. Defense Mechanisms: A discussion on potential defense mechanisms or mitigations would make the work have better impacts in this area and further highlight the significance of the threat.

2. Assumptions: In the experiments, the proportion of malicious clients is 1/5, which might not be realistic in real practice.

3. More Experiments: The attack is evaluated on two datasets (CIFAR10 and CelebA). Expanding the evaluation to include more diverse datasets could further support the work.

**Questions:**

NA

**Limitations:**

The authors discussed the limitations and potential negative societal impact of the work in Appendix.

---

> ### Author Rebuttal · Authors · 2024-08-07
>
> Thank you for your insightful comments and high appreciation of the innovation and experiments in our work. We address your concerns below:
>
> **Response to W1:**
>
> Thank you for your suggestion. Here is our discussion on the defense mechanisms:
>
> -	According to Table 1 and Fig.3 in the main paper, Multi-Krum shows the best defense performance, followed by RFA. Under the Multi-Krum defense, no attack strategy can rapidly compromise the global model to achieve MSE below 0.1 within the designated round. This is mainly because Multi-Krum is effective at detecting malicious updates. When the attacker uses a small scale to bypass Multi-Krum, the defense mechanism can still dilute the malicious update by weighted averaging the remaining updates. Although the experiment in Appendix A.1 shows that the defense is not reliable with longer training, Multi-Krum could be a good start for future work.
> -	Additionally, the differential privacy algorithm can render indicators invalid while sacrificing generative performance, as discussed in Appendix F (Lines 666-671). This is mainly because noise or norm clipping treats all parameters uniformly. Since AdaSCP requires specific indicators with a large magnification factor, locating candidate indicators and filtering outliers after comparison with other updates would be more efficient and effective. This method is preferable to comparing the distance of all parameters, which tends to average out outliers and makes it difficult to filter out malicious clients.
> -	Moreover, the experiment in Appendix A.2 shows that our backdoors diminish after 100 rounds of continued training with clean data, suggesting a potential mitigation strategy for future work. Lastly, since the triggers are implanted in the input noise, releasing only the generated results without exposing the model parameters is another viable defense.
>
> We will add a new section to discuss the potential defense mechanisms and mitigations of *DataStealing* backdoors.
>
> **Response to W2:**
>
> We have conducted experiments with more clients, such as 1/8 and 1/10, as detailed in Appendix A.6. These results show that AdaSCP remains effective in executing attacks under more clients. For experiments with more attackers, we tested with two malicious clients, each having 500 target images. The results for 2/5 are shown below:
>
> Table B: *DataStealing* with two attackers applying AdaSCP on CIFAR10 under Non-IID distribution.
> | Defenses | FID | $\text{MSE}_1$ | $\text{MSE}_2$ | $\text{MSE}_{mean}$ |
> |:----------|:-------|:-------|:-------|:-------|
> | Multi-Krum |14.97|0.0886|0.0898|0.0892|
> |Foolsgold|33.24|0.0174|0.0229|0.0201|
> | Multi-metrics |15.61|0.0324|0.0608|0.0466|
>
> Table B shows that AdaSCP can be applied in scenarios with multiple attackers. However, the attackers may interfere with each other, leading to varying attacking performances. Additionally, having two attackers causes the FID to rise, mainly because they disrupt normal training with scaled updates. As a result, the magnitude of the weighted average updates is double that of a single attacker. This issue could be mitigated by dividing the updates by the number of attackers. We leave this for future research.
>
> **Response to W3:**
>
> Thank you for your suggestion. We conducted an additional experiment with the LSUN bedroom dataset, which has a higher resolution of 256x256. More details can be found in our global response and the PDF file.

---

### Official Review · Reviewer_CPaA · 2024-07-13

**Soundness:** 3
**Presentation:** 3
**Contribution:** 3
**Rating:** 6
**Confidence:** 4

**Summary:**

The paper titled "DataStealing: Steal Data from Diffusion Models in Federated Learning with Multiple Trojans" investigates a novel privacy vulnerability in federated learning (FL) when training diffusion models. The authors introduce a new attack methodology named DataStealing, which leverages multiple Trojans to exfiltrate private data from local clients. The attack utilizes Combinatorial Triggers (ComboTs) to map extensive data and proposes the Adaptive Scale Critical Parameters (AdaSCP) attack to bypass advanced FL defenses.

**Key Contributions:**
* 1\. **Identification of Vulnerability**:

    - The paper identifies a new privacy threat in FL where diffusion models can leak substantial private data. Despite stringent privacy measures, attackers can exploit these models to steal high-quality local data.

* 2\. **Combinatorial Triggers (ComboTs)**:

    - A method to select multiple triggers for backdoor attacks, significantly increasing the capability to map and steal large amounts of private data.

* 3\. **Adaptive Scale Critical Parameters (AdaSCP)**:

    - An attack strategy that evaluates and scales critical parameter updates, making the malicious updates indistinguishable from benign ones. This circumvents distance-based defenses effectively.

* 4\. **Experimental Validation**:

    - Extensive experiments demonstrate the ability of the proposed methods to leak thousands of images from training diffusion models in an FL setup. AdaSCP is shown to be highly effective against advanced FL defenses.

**Conclusion**:

The paper highlights the severe privacy risks associated with training diffusion models in FL and calls for more robust defensive measures to protect against such vulnerabilities. The findings emphasize the need for continuous advancements in FL security to safeguard private data.

**Strengths:**

**Strengths of the Paper**
**Originality**
- **New Vulnerability Identification**: The paper identifies a previously unrecognized vulnerability in federated learning (FL) when using diffusion models, termed as DataStealing. This identification highlights a novel attack surface in the intersection of generative models and FL.
- **Innovative Attack Methodologies**: The introduction of Combinatorial Triggers (ComboTs) and the Adaptive Scale Critical Parameters (AdaSCP) attack represents significant originality. ComboTs utilize multiple triggers to enhance the attack’s capability, while AdaSCP circumvents advanced defenses by adaptively scaling critical parameter updates.
- **Creative Combination of Techniques**: The paper creatively combines the concepts of Trojans in generative models with adaptive scaling techniques to formulate an effective attack strategy against FL systems.

**Quality**

- **Rigorous Experimental Validation**: The paper presents extensive experiments to validate the effectiveness of the proposed attack methodologies. The results are comprehensive, demonstrating the ability of AdaSCP to defeat various advanced FL defenses.

- **Detailed Methodological Explanation**: The methodologies are described in detail, with clear explanations of the underlying principles and the implementation of the attack strategies. This thorough presentation ensures that the proposed methods are reproducible and verifiable.

- **Robustness of Results**: The experimental results are robust, showing consistent performance across different datasets (CIFAR10 and CelebA) and FL settings (IID and non-IID distributions).

**Clarity**

- **Well-Organized Structure**: The paper is well-organized, guiding the reader logically from the identification of the problem to the proposed solutions and experimental validation.

- **Clear Writing Style**: The writing is clear and concise, making complex technical content accessible. Each section is well-articulated, with smooth transitions that maintain the reader’s engagement.

- **Effective Use of Visual Aids**: Figures and tables are used effectively to illustrate key points and results. These visual aids enhance the reader’s understanding of the methodologies and the significance of the findings.

**Significance**

- **High Relevance to Privacy and Security**: The paper addresses a critical issue in the domain of FL, highlighting significant privacy risks associated with training diffusion models. The identified vulnerabilities and proposed solutions are highly relevant to the ongoing discussions about data privacy and security in machine learning.

- **Impact on Future Research**: The findings of this paper are likely to stimulate further research in the field, prompting the development of more robust defense mechanisms in FL. By uncovering new attack strategies, the paper sets the stage for advancements in both attack and defense techniques.

- **Broad Applicability**: The proposed methodologies and findings are not limited to a specific application but are broadly applicable to various domains where FL and generative models are used. This broad applicability enhances the overall significance of the paper.

**Conclusion**

The paper "DataStealing: Steal Data from Diffusion Models in Federated Learning with Multiple Trojans" excels in originality, quality, clarity, and significance. It introduces novel vulnerabilities and attack methodologies in FL, backed by rigorous experimental validation and clear presentation. The contributions of this paper are highly relevant to the research community and hold significant potential for future advancements in privacy and security in machine learning.

**Weaknesses:**

**Weaknesses of the Paper**

**Limited Discussion on Defense Mechanisms**

- **Lack of Proactive Defense Strategies**: While the paper thoroughly explores the vulnerabilities and proposes robust attack methodologies, it falls short in discussing proactive defense mechanisms. It would significantly benefit from suggesting or exploring potential defense strategies to mitigate the identified vulnerabilities. Including a section on how to strengthen FL systems against such attacks would provide a more balanced view.

**Depth of Contextualization**
- **Comparative Analysis with Prior Work**: Although the paper positions its contributions within the broader context of federated learning and privacy security, a deeper comparative analysis with specific prior studies would enhance its impact. For example, more detailed comparisons with existing backdoor and Trojan attack methodologies would help in highlighting the advancements made by this work.
- **Exploration of Related Work**: The related work section could be expanded to include a more comprehensive review of similar vulnerabilities in federated learning and generative models. This would help in better contextualizing the novelty and significance of the proposed methods.

**Experimental Scope**
- **Scalability and Generalizability**: While the experiments are robust, they are limited to specific datasets (CIFAR10 and CelebA) and certain configurations (IID and non-IID distributions). Exploring the scalability and generalizability of the proposed attack methodologies to other datasets and real-world scenarios would strengthen the empirical validation. For instance, including more diverse datasets or different FL frameworks could provide insights into the broader applicability of the findings.

- **Extended Analysis of Parameters**: The paper briefly touches upon the impact of various parameters, such as the proportion of critical parameters and patch sizes in ComboTs. A more detailed and systematic analysis of how these parameters influence the attack's effectiveness across different settings would be beneficial. Including ablation studies on a wider range of parameters and configurations would offer a deeper understanding of the methods' robustness.

**Presentation Improvements**

- **Simplifying Technical Explanations**: Some sections of the paper, particularly those explaining the technical details of the methodologies, could be simplified for better readability. This would make the paper more accessible to a broader audience, including those who may not have a deep technical background in the specific area.

- **Enhanced Flow and Readability**: Improving the flow and readability of certain sections, particularly the methodology and experiment results, would enhance the overall presentation. Ensuring smoother transitions between sections and sub-sections would maintain the reader’s engagement and comprehension.

**Specific Suggestions for Improvement**

* 1\. **Incorporate Proactive Defense Mechanisms**: Add a section discussing potential defense strategies to mitigate the vulnerabilities identified. This could include theoretical approaches, proposed defense mechanisms, or an exploration of existing techniques that could be adapted.
* 2\. **Expand Comparative Analysis**: Deepen the comparative analysis with prior work, providing detailed discussions on how the proposed methods advance the state-of-the-art in backdoor and Trojan attacks in FL.
* 3\. **Broaden Experimental Validation**: Expand the experimental scope to include more diverse datasets and configurations. Conduct a systematic analysis of various parameters influencing the attack's effectiveness, with detailed ablation studies.
* 4\. **Simplify and Enhance Readability**: Simplify technical explanations where possible, and improve the overall flow and readability of the paper. This could involve rephrasing complex sections and ensuring smooth transitions between different parts of the paper.

By addressing these weaknesses, the paper can further solidify its contributions and provide a more comprehensive and impactful addition to the research area.

**Questions:**

**Questions and Suggestions for the Authors**

**Questions**

* 1\. **Defense Mechanisms**:

    - What proactive defense strategies do you suggest to mitigate the vulnerabilities identified in the paper? Have you considered evaluating existing defense mechanisms or proposing new ones specifically tailored to counter the DataStealing attack?

* 2\. **Comparative Analysis**:

    - Can you provide a more detailed comparative analysis with specific prior studies on backdoor and Trojan attacks in federated learning? How do your proposed methods significantly advance the state-of-the-art compared to these existing approaches?

* 3\. **Scalability and Generalizability**:

    - How do you anticipate your proposed attack methodologies (ComboTs and AdaSCP) will perform on more diverse datasets and in real-world scenarios? Have you considered evaluating your methods on other datasets or federated learning frameworks to validate their scalability and generalizability?

* 4\. **Parameter Sensitivity**:

    - Can you provide a more systematic analysis of how different parameters (e.g., proportion of critical parameters, patch sizes in ComboTs) influence the effectiveness of the attacks? Detailed ablation studies on these parameters would be beneficial for understanding the robustness of your methods.

* 5\. **Complexity and Computation**:

    - What are the computational costs and complexities associated with implementing your proposed methods (ComboTs and AdaSCP)? How feasible is it to execute these attacks in practical settings, considering the required computational resources and time?

**Suggestions**

* 1\. **Proactive Defense Strategies**:

    - Include a dedicated section discussing potential defense strategies to mitigate the vulnerabilities identified. This could involve theoretical approaches, proposed defense mechanisms, or an exploration of existing techniques that could be adapted to counter DataStealing.

* 2\. **Enhanced Comparative Analysis**:

    - Deepen the comparative analysis with prior work, providing detailed discussions on how your proposed methods advance the state-of-the-art in backdoor and Trojan attacks in federated learning. This can help in better contextualizing the novelty and significance of your contributions.

* 3\. **Broaden Experimental Validation**:

    - Expand the experimental scope to include more diverse datasets and configurations. Conduct a systematic analysis of various parameters influencing the attack's effectiveness, with detailed ablation studies. This would strengthen the empirical validation and demonstrate the broader applicability of your findings.

* 4\. **Simplify Technical Explanations**:

    - Simplify the technical explanations where possible to make the paper more accessible to a broader audience, including those without a deep technical background in the specific area. This would enhance the overall readability and impact of the paper.

* 5\. **Detailed Discussion on Limitations**:

    - Include a more detailed discussion on the limitations of your proposed methods. Address potential challenges and constraints in implementing the attacks, and suggest future research directions to overcome these limitations.

By addressing these questions and suggestions, the authors can provide a more comprehensive and impactful contribution to the research area, enhancing the clarity, robustness, and significance of their work.

**Limitations:**

**Assessment of Limitations and Potential Negative Societal Impact**

**Addressed Limitations**

* 1\. **Technical Limitations**:

    - The authors briefly mention the limitations related to the computational complexity and scalability of their proposed methods. They acknowledge the challenges in training diffusion models with ComboTs and the sensitivity of these models to gradient updates.

* 2\. **Experimental Constraints**:

    - The paper indicates that the experiments were conducted on specific datasets (CIFAR10 and CelebA) and acknowledges the need for further validation on more diverse datasets and real-world scenarios.

* 3\. **Defensive Measures**:

    - There is a mention of the need for further research to develop robust defensive mechanisms against the identified vulnerabilities in federated learning, suggesting an awareness of the current gap in defensive strategies.

**Potential Negative Societal Impact**

* 1\. **Privacy and Security Risks**:

    - The paper highlights significant privacy risks associated with federated learning, especially when training diffusion models. The authors acknowledge the potential misuse of the proposed attack methodologies to steal private data, underscoring the importance of addressing these vulnerabilities to protect user privacy.

**Constructive Suggestions for Improvement**

* 1\. **Detailed Discussion on Limitations**:

    - The authors should provide a more comprehensive discussion on the limitations of their proposed methods. This could include:

        - A deeper exploration of the computational costs and complexities associated with implementing ComboTs and AdaSCP.

        - An assessment of the scalability and generalizability of the proposed methods across different datasets and federated learning frameworks.

        - Potential challenges in deploying these attacks in real-world scenarios, considering practical constraints and resource limitations.

* 2\. **Proactive Defense Strategies**:

    - Include suggestions or preliminary evaluations of potential defense mechanisms to counter the DataStealing attack. This would provide a more balanced view and contribute to advancing the field by not only identifying vulnerabilities but also proposing ways to mitigate them.

* 3\. **Ethical Considerations and Responsible Disclosure**:

    - The authors should discuss the ethical considerations and responsible disclosure practices followed in conducting this research. This could include:
        - Steps taken to ensure that the research does not cause harm.
        - Collaboration with organizations or communities to address the identified vulnerabilities responsibly.
        - Suggestions for policy or regulatory measures to protect against the misuse of such attack methodologies.

* 4\. **Broader Societal Impact**:

    - Expand on the broader societal impact of the research by discussing:
        - How the findings could influence the development of privacy-preserving technologies in federated learning.
        - The implications for industries and applications relying on federated learning, such as healthcare, finance, and IoT.
        - Potential benefits of raising awareness about these vulnerabilities to foster more robust and secure machine learning practices.

**Conclusion**

The authors have made a good start in addressing the limitations and potential negative societal impacts of their work. By providing a more detailed discussion on these aspects and incorporating the constructive suggestions above, the authors can enhance the comprehensiveness and responsibility of their research. This approach would contribute positively to the field and ensure that the research is conducted and presented in an ethically sound and impactful manner.

---

> ### Author Rebuttal · Authors · 2024-08-07
>
> Thank you for your thorough review and appreciation of the novelty and experiments of our work. We address your concerns below:
>
> **Response to "Discussion on Defense Mechanisms"**:
>
> Thank you for your suggestion. Here is our discussion on the defense mechanisms:
>
> -	According to Table 1 and Fig.3 in the main paper, Multi-Krum shows the best defense performance, followed by RFA. Under the Multi-Krum defense, no attack strategy can rapidly compromise the global model to achieve MSE below 0.1 within the designated round. This is mainly because Multi-Krum is effective at detecting malicious updates. When the attacker uses a small scale to bypass Multi-Krum, the defense mechanism can still dilute the malicious update by weighted averaging the remaining updates. Although the experiment in Appendix A.1 shows that the defense is not reliable with longer training, Multi-Krum could be a good start for future work.
> -	Additionally, the differential privacy algorithm can render indicators invalid while sacrificing generative performance, as discussed in Appendix F (Lines 666-671). This is mainly because noise or norm clipping treats all parameters uniformly. Since AdaSCP requires specific indicators with a large magnification factor, locating candidate indicators and filtering outliers after comparison with other updates would be more efficient and effective. This method is preferable to comparing the distance of all parameters, which tends to average out outliers and makes it difficult to filter out malicious clients.
> -	Moreover, the experiment in Appendix A.2 shows that our backdoors diminish after 100 rounds of continued training with clean data, suggesting a potential mitigation strategy for future work. Lastly, since the triggers are implanted in the input noise, releasing only the generated results without exposing the model parameters is another viable defense.
>
> We will add a new section to discuss the potential defense mechanisms and mitigations of *DataStealing* backdoors.
>
> **Response to "Comparative Analysis":**
>
> Thank you for your suggestion. We have discussed AdaSCP in comparison with prior attack methods in Section 4.1 (Lines 263-281). In summary, Data Poison and PGD Poison encounter difficulties in effectively implanting multiple backdoors into diffusion models. Model Poison fails to overcome advanced FL defenses due to improper scale values. BC Layer Substitution is unsuitable for training diffusion models and results in training collapse. Our AdaSCP outperforms other methods by using critical parameters with adaptive scale factors, which balance stealth and efficiency and prevent training collapse in diffusion models. We will add a conclusion to Section 4.1 to help readers understand the advantages of our method.
>
> **Response to "Scalability and Generalizability":**
>
> Thank you for your suggestion. We conducted an additional experiment with the LSUN bedroom dataset, which has a higher resolution of 256x256. More details can be found in our global response and the PDF file.
>
> **Response to "Extended Analysis of Parameters":**
>
> Thank you for your suggestion. We will expand the analysis of the impact of various parameters in Appendix A. As for the proportion of critical parameters, the results in Appendix A.3 show that 0.4 is the trade-off point between the performance and critical parameter count. As for the patch size of ComboTs, the results in Appendix A.5 indicate that a 3x3 patch achieves the best performance. Smaller patches are insufficient to distinguish the triggers from the noise, while larger patches obscure the image and degrade performance. We will add more details to enhance the parameter analysis.
>
> **Response to "Complexity and Computation":**
>
> The complexity of ComboTs depends on the time of selecting the triggers from potential positions, as defined in Section 3.2 (Lines 123-125). For AdaSCP, the efficiency is influenced by the batch size and a hyperparameter, threshold $\mathcal{T}$, demonstrated in Appendix C.1 and Algorithm 4. The complexity primarily arises from the process of searching for critical parameters and identifying candidate indicators. The time consumption of AdaSCP is various with different hyperparameters and the model complexity. For example, when finetuning with the LSUN bedroom dataset, the resolution is 256x256, the batch size is 3 and the hyperparameter $\mathcal{T}$ is 0.05, the running time for finding critical parameters and candidate indicators is 2 minutes and 29 seconds. This process is conducted after all candidate indicators have been exhausted, as demonstrated in Algorithm 1. In our experiments, we set the number of candidate indicators to 10. Enlarging the number of candidate indicators is a way to increase the efficiency of our method.
>
> We will add more details for the complexity analysis in the revised version.
>
> **Response to "Presentation Improvements":**
>
> Thank you for your suggestion. To enhance the readability of our paper, we will polish the entire manuscript and move some details to the Appendix.
>
> **Response to "Limitations":**
>
> As discussed above, we will include an additional section on the proactive defenses against *DataStealing*. Due to ethical considerations and responsible disclosure, we will assist researchers and managers in preventing the exploitation of this vulnerability. Raising awareness about this issue can lead to the adoption of stricter security standards in the design and implementation of training diffusion models with FL, thereby minimizing the risks of data breaches and misuse. Additionally, we will release our code to encourage the development of more advanced privacy-preserving mechanisms.
>
> Furthermore, we will expand the discussion to cover the limitations and potential societal impacts on privacy-preserving technologies in FL and relevant industries.

---

> > ### Comment · Reviewer_CPaA · 2024-08-12
> >
> > We appreciate the authors' patience and thorough rebuttal. After careful consideration and review, we decide to maintain the original score.

---

### Official Review · Reviewer_H1Y5 · 2024-07-13

**Soundness:** 2
**Presentation:** 3
**Contribution:** 2
**Rating:** 6
**Confidence:** 3

**Summary:**

This paper studies the privacy risks of diffusion models in federated learning (FL) through the lens of data stealing attacks. The authors propose the ComboTs method to target a large amount of images. To further defeat advanced distance-based FL defenses, the authors propose the AdaSCP attack method which adaptively scales the critical parameters in the updating process. The evaluation results demonstrate the effectiveness of this attack.

**Strengths:**

- The first work to consider the privacy risks of diffusion models under the FL setting
- Good performance with low MSE of the recovered images

**Weaknesses:**

- Problematic attack settings
- Insufficient experiments and explanations

**Questions:**

This paper explores the data leakage risks of diffusion models in federated learning. The authors conduct the data stealing attack via backdoor attacks. To target a large amount of images and bypass the advanced FL defenses, the authors propose the ComboTs and AdaSCP methods, respectively. The evaluations showcase the effectiveness of the proposed data stealing attack.

However, there are still some critical issues.

- The attack settings are problematic. In this paper, the attacker is assumed to have access to the training dataset of the poisoned client, such that the attacker can insert the backdoor triggers into the target images, and finally the attacker will be able to recover the target images according to their backdoor trigger patterns. However, since the attacker already has access to the target images, why bother stealing them using the proposed complex methods? Just because of the so-called "Under strict data privacy protection, attackers can not upload or download images from the infiltrated client" in Section 3.1? This assumption does NOT make any sense. The attacker can even check every pixel value of the target image to locally reconstruct it without downloading it from the client. I would suggest the authors clearly re-explain the motivation and the settings of the attack.

- The authors mentioned the FID metric to measure the performance of diffusion models. However, the authors did not clearly claim either higher or lower FID is better. Furthermore, when discussing the evaluation results, the authors only focused on the MSE score. I would suggest the authors elaborate more on this metric if they use it as an evaluation metric.

- The authors only conduct the experiments on two image datasets with low resolutions. I would suggest the authors also evaluate the performance of their attack on another dataset with a much higher resolution, e.g., a subset of ImageNet.

**Limitations:**

See my detailed questions

---

> ### Author Rebuttal · Authors · 2024-08-06
>
> Thank you for your constructive comments. We address your concerns below:
>
> **Response to Q1:**
> 1) **Access to data does not mean it can be stolen.**
>
> - Federated Learning (FL) aims to protect data privacy by only sending model updates to the central server. In the FL setting, private data **cannot** leave the client. The assumption that an attacker can check every pixel value to locally reconstruct target images conflicts with the federated learning framework, where only model updates are transmitted, not raw data.
> - FL is often used to train models on highly sensitive or valuable data, such as medical records. In the era of deep learning, data is a critical asset and involves organizational security issues. According to *Article 5 and Article 32 of the European Union General Data Protection Regulation (GDPR)*[1], organizations are required to take strict measures to protect data privacy and security. For such highly protected data, some organizations enforce strict access controls, including banning USB drives, restricting network access, and disabling copy-paste, ensuring data cannot be downloaded from the client. In this environment, sending unauthorized messages would trigger security alarms. However, model updates are authorized for transfer in the FL system. Thus, it is practical that attackers can infiltrate the training process and implant backdoors to steal data but do not have the ability to extract data straightforwardly in the FL framework. The malicious goal can be secretly achieved by modifying the training code of the client without any risky transmission.
>
> 2) **An example to enhance FL trustworthiness and security.**
>
> - People might assume that advanced federated learning frameworks are sufficiently secure and overlook this potential danger. However, a false sense of security can be more dangerous and may be used for malicious purposes, such as espionage. Our work is the first to demonstrate that training diffusion models under the FL setting poses a threat of leaking thousands of images by implanting multiple backdoors, which control the generative behavior by only uploading model updates in FL. Our aim is to enhance the trust and security of FL and protect data privacy in the end.
>
> We will add more key details to the introduction to ensure that readers can more easily understand our setting in the revised manuscript.
>
> [1] Voigt, P., & Von dem Bussche, A. (2017). The eu general data protection regulation (gdpr). *A Practical Guide, 1st Ed., Cham: Springer International Publishing, 10*(3152676), 10-5555.
>
> **Response to Q2:**
>
> Thank you for your suggestion. In Table 1, we have indicated that "&darr;" means lower is better. The FID (Fréchet Inception Distance) evaluates the quality of generated images by calculating the Fréchet distance between the feature distributions of generated and real images in a pre-trained Inception network. A lower FID indicates that the generated images are closer to real images, implying higher quality. The FID becomes very high when the diffusion model experiences training collapse, which mainly occurs in Model Poison and BC Layer Substitution due to improper scale rates and training approaches (Lines 278-281). As for AdaSCP, successful multiple backdoor attacks result in a slight increase in the FID score, averaging +10.31 on CIFAR-10 and +1.73 on CelebA compared to the pre-trained diffusion model, as demonstrated in Appendix F (Lines 663-666). This effect can be alleviated with longer training, as shown in Appendix A.1 (Lines 503-505). Since FID cannot directly reflect the performance of implanted backdoors, we primarily focus on comparing the MSE metric. More details about FID are provided in the Appendix. In the revised version of the main paper, we will add more details to help readers better understand the FID metric and the table results.
>
> **Response to Q3:**
>
> Thank you for your suggestion. We conducted an additional experiment with the LSUN bedroom dataset, which has a higher resolution of 256x256. More details can be found in our global response and the PDF file.

---

### Author Rebuttal · Authors · 2024-08-07

First of all, we would like to thank you for your time, constructive critiques, and valuable suggestions, which have greatly helped us improve our work. We are also grateful that the reviewers unanimously regard our work as novel and convincing. Below, we respond to the suggestions regarding experiments on additional datasets with higher resolution.

We adopted the pretrained model provided by the official DDPM implementation, which is trained on the LSUN bedroom dataset with a 256x256 resolution. The *DataStealing* experiment is conducted by fine-tuning the pretrained model with a subset of the LSUN bedroom dataset, containing approximately 300,000 images.

Considering the limited rebuttal time and the computational resources required for training, we set the batch size to 8 and the number of target images to 50. Each client trains on 500 images for one round. The EMA scale is decreased from 0.9999 to 0.999 for quicker convergence. The quantitative and qualitative results are shown in Table A and Figure A in our uploaded PDF file. We compare AdaSCP with several effective attack methods, such as Data Poison, Model Poison, and PGD Poison. The results show that AdaSCP still outperforms other methods in the *DataStealing* task under advanced FL defenses, further supporting our work.

To address the lengthy 20-hour FID calculation time required for sampling 50,000 images at a 256x256 resolution, we used only 5,000 images, reducing the inference time to 2 hours. However, the limited training and inference time lead to higher FID values compared to other datasets. We will continue working on the LSUN bedroom dataset to provide more complete results in the revised manuscript.

Thank you again for your valuable time. We sincerely look forward to further discussions with you.

---

### Decision · Program_Chairs · 2024-09-25

**Decision:**

Accept (poster)

**Comment:**

The paper identifies a new threat of data extraction from diffusion models trained with federated learning.  The combinatorial trigger methodology was appreciated by the reviewers as a creative design, as was the use of adaptive scale critical parameters. The paper was clearly written, and the experiments were overall convincing to the reviewers (though experiments can always be expanded). Please do make sure to include your updated results on the LSUN bedrooms dataset. On my own reading of the paper, I was curious how the method works against other distance-based defenses tailored to FL, such as Wang et al "Towards a defense against federated backdoor attacks under continuous training". Overall, the paper presents a solid step in deepening our understanding of adversarial attacks in the federated setting, particularly for diffusion models.